# Geometry-Aware Neural Optimizer for Shape Optimization and Inversion

Guoze Sun [* 1]   Tianya Miao [* 1]   Haoyang Huang [1]   Huaguan Chen [1]   Han Wan [1]   Rui Zhang [1]   Hao Sun [1]

## Abstract

Geometry is central to PDE-governed systems, motivating shape optimization and inversion. Classical pipelines conduct costly forward simulation with geometry processing, requiring substantial expert effort. Neural surrogates accelerate forward analysis but do not close the loop because gradients from objectives to geometry are often unavailable. Existing differentiable methods either rely on restrictive parameterizations or unstable latent optimization driven by scalar objectives, limiting interpretability and part-wise control. To address these challenges, we propose Geometry-Aware Neural Optimizer (**GANO**), an end-to-end differentiable framework that unifies geometry representation, field-level prediction, and automated optimization/inversion in a single latent-space loop. GANO encodes shapes with an auto-decoder and stabilizes latent updates via a denoising mechanism, and a geometry-informed surrogate provides a reliable gradient pathway for geometry updates. Moreover, GANO supports part-wise control through null-space projection and uses remeshing-free projection to accelerate geometry processing. We further prove that denoising induces an implicit Jacobian regularization that reduces decoder sensitivity, yielding controlled deformations. Experiments on three benchmarks spanning 2D Helmholtz, 2D airfoil, and 3D vehicles show state-of-the-art accuracy and stable, controllable updates, achieving up to $+55.9\%$ lift-to-drag improvement for airfoils and $\sim 7\%$ drag reduction for vehicles.

## 1. Introduction

*Geometry* is one of the most important control variables for PDE-governed systems (Yau, 1982). This motivates two

---
[*]Equal contribution [1]Gaoling School of Artificial Intelligence, Renmin University of China. Correspondence to: Rui Zhang <rayzhang@ruc.edu.cn>, Hao Sun <haosun@ruc.edu.cn>.

*Proceedings of the $43^{rd}$ International Conference on Machine Learning*, Seoul, South Korea. PMLR 306, 2026. Copyright 2026 by the author(s).

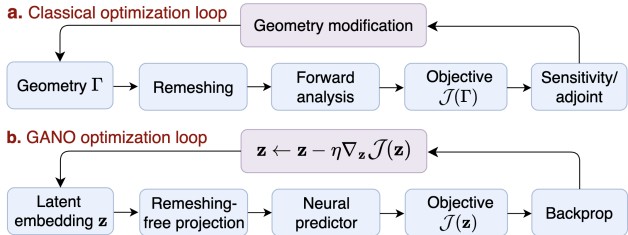

*Figure 1.* Comparison between a classical geometry optimization loop (**a**) and the proposed GANO loop (**b**).

key tasks: *optimization* (Fei et al., 2025), which designs geometries to meet objectives under manufacturing constraints, and *inversion* (Bastek & Kochmann, 2023; Salo, 2024), which infers unknown shapes from observations. Efficient and reliable geometric optimization/inversion reduces computational cost and is crucial for manufacturing. Classical geometry optimization and inversion (Jameson, 2003; Peter & Dwight, 2010) follow an iterative workflow that includes forward analysis, geometry modification, and remeshing (Fig. 1**a**). Therefore, forward solves and geometry processing become computational bottlenecks, leading to slow iteration and heavy expert intervention.

To reduce the cost of the forward step, recent advances have enabled fast neural surrogates for PDE-governed systems with complex geometries (Li et al., 2023; Wu et al., 2024a; Zeng et al., 2025). These models can predict solution fields directly on irregular meshes or point clouds, replacing expensive numerical solvers with faster inference. However, accelerating the forward pass alone does not yield end-to-end geometry optimization. In practice, geometry updates are typically performed through non-differentiable operations (e.g., CAD edits and remeshing), which significantly limit optimization speed and require strong specialized knowledge and experience.

Beyond accelerating forward analysis, deep learning has also been leveraged for geometry optimization and inversion. For geometric representation, existing approaches can be divided into *parametric* and *latent-space* methods. *Parametric methods* optimize over human-designed, low-dimensional parameterizations (e.g., design-variable formulations), often coupled with learned surrogates or learning-guided search (Allen et al., 2022; Shukla et al., 2024; Rehmann et al., 2025). Their limited design spaces yield stable, in-

terpretable updates, but restrict expressivity and coverage of complex 3D geometries. In contrast, *latent-space methods* (Vatani et al., 2025; Hao et al., 2025) learn encoder–decoder representations and optimize directly in the latent space, offering richer geometric variability. Yet gradient-based latent updates can easily drift off the data-supported shape manifold because the latent space is high-dimensional and weakly constrained, resulting in geometries catering to optimization targets but impractical in reality (Wu et al., 2024b). Moreover, many pipelines predict only scalar objectives and optimize them directly (Tran et al., 2024; Vatani et al., 2025), rather than operating on field-level predictions, which weakens interpretability and fine-scale refinement.

To address the lack of industrial-grade end-to-end automation in geometry optimization, we propose the **G**eometry-**A**ware **N**eural **O**ptimizer (**GANO**), an end-to-end differentiable framework that unifies *representation*, *prediction*, and *optimization*, yielding results with excellent performance improvements as well as practicality for industrial applications. GANO replaces the classical loop with an automated latent-space procedure that enables accurate prediction, stable geometry updates, and part-wise control (Fig. 1**b**). For *representation*, GANO encodes each shape into a latent code **z** with an auto-decoder STABLESDF, and uses a denoising technique to stabilize gradient-based optimization and inversion. For *forward prediction*, GANO incorporates the TRANSOLVER (Wu et al., 2024a) backbone with an efficient geometry-informed mechanism, providing an informative gradient pathway from objectives to **z**. For *optimization* and *inversion*, GANO optimizes geometry by updating **z** while supporting variable, region-aware objectives, and part-wise constraints through null-space projection (e.g., freezing specified components and optimizing others). Moreover, we maintain boundary-consistent queries onto the updated surface via remeshing-free projection.

We evaluate GANO on three challenging benchmarks, including forward modeling and inversion for the Helmholtz equation, 2D airfoil flow prediction and lift-maximizing optimization, and 3D vehicle surface-pressure prediction and drag-minimizing optimization. Across all tasks, GANO achieves state-of-the-art performance and enables stable, remeshing-free geometry updates with part-wise control. Overall, our contributions can be summarized as follows:

1. We propose **GANO**, an end-to-end differentiable framework for industrial-grade geometry optimization, which unifies *geometry representation*, *field-level prediction*, and *automated optimization/inversion* in a single loop.

2. We prove that the denoising mechanism in the geometry representation induces Jacobian regularization that reduces the decoder's sensitivity to latent perturbations and yields controlled shape deformations during optimization.

3. GANO achieves consistent state-of-the-art performance across three challenging benchmarks. Specifically, it achieves up to **+55.9%** improvement in the lift-to-drag ratio (2D airfoil) and up to ~**7%** reduction in drag (3D vehicle).

## 2. Related Works

**Neural PDE Solvers.** Neural PDE solvers are used to accelerate the forward computation for PDE-governed systems, achieving fast inference once trained. Representative works include FNO (Li et al., 2021; Hao et al., 2024), DEEPONET (Lu et al., 2019; Karumuri et al., 2026) and other variants (Zhang et al., 2024; Hu et al., 2025; Bhaganagar & Chambers, 2025). To handle complex geometries, recent work extends neural PDE solvers to irregular discretization settings. Graph-based surrogate models PDE states on unstructured meshes, allowing direct learning on domains with complex boundaries (Zeng et al., 2025; Liu et al., 2025). In parallel, transformer-style architectures treat point samples as tokens and capture long-range interactions with attention (Hao et al., 2023; Wu et al., 2024a; Luo et al., 2025). Geometry-informed operators map irregular geometries to a regular grid, enabling efficient inference in the latent space (Li et al., 2023; Liu et al., 2026). Furthermore, several approaches aim to accelerate the forward analysis of aerodynamics (Chen et al., 2025; Liu & Chen, 2025; Gu et al., 2026). Despite these advances, most neural PDE solvers focus solely on accelerating the forward pass. For shape optimization/inversion, the geometry is updated across iterations, which is non-differentiable and disrupts gradient propagation to the shape.

**Neural Optimizer.** Prior work has explored different ways to achieve neural shape optimization (Li et al., 2022). Parametric methods optimize geometry in a hand-crafted parameter space by predicting physical quantities and updating the design variables via backpropagation (Sun & Wang, 2019; Allen et al., 2022; Shukla et al., 2024; Rehmann et al., 2025). While effective on structured designs, this route is constrained by limited parametrization and struggles with complex objects like vehicles. To enlarge design space, several methods perform gradient-based optimization directly in a learned *latent* space (Gómez-Bombarelli et al., 2018; Mescheder et al., 2019). Typical pipelines encode shapes, learn a surrogate for optimization objectives (e.g., drag), and iteratively update the latent code via gradients (Tran et al., 2024). Some variants fine-tune parts of the encoder to modify the geometry-latent mapping (Vatani et al., 2025). Despite improved flexibility, voxel/occupancy representations often sacrifice geometric fidelity. Recently, generative approaches embed optimization into diffusion or flow matching by injecting objective gradients into the sampling dynamics (Hao et al., 2025; You et al., 2025), which still rely on decoded-space refinement and increase computation

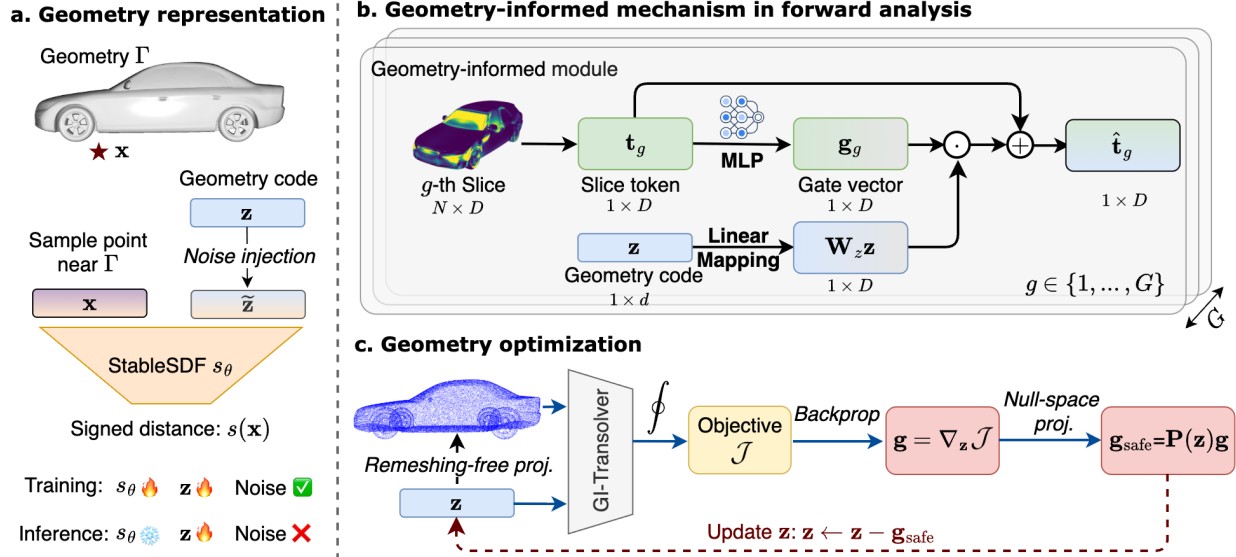

*Figure 2.* **Pipeline of GANO**. **a**. *Geometry representation:* GANO models $\Gamma$ as an implicit SDF $s_\theta(\mathbf{x})$ and uses denoising-style augmentation during training. **b**. *Geometry-informed mechanism in forward analysis:* A geometry-informed module to inject geometry information and creates a gradient pathway. **c**. *Optimization:* We backpropagate $\nabla_\mathbf{z}\mathcal{J}$ and apply a null-space projection $P(\mathbf{z})$ to achieve controllable geometry update.

overhead. Moreover, (Chen et al., 2026) depends only on scalar prediction and lacks the ability of part-wise optimization. We provide a more detailed discussion in Appendix B.

**Implicit Neural Representations.** Implicit neural representations (INRs) represent continuous signals by mapping spatial or spatio-temporal coordinates to signal values with neural networks. They have been widely used for geometry, vision, and physical-field modeling because they provide a resolution-independent and differentiable representation of continuous domains. For geometry modeling, DeepSDF (Park et al., 2019) learns a continuous signed distance function conditioned on a latent code, enabling compact shape representation, interpolation, and reconstruction. Beyond conventional MLPs, SIREN (Sitzmann et al., 2020) introduces sinusoidal activation functions to improve the representation of high-frequency signals and their derivatives, which is particularly useful for physical signals governed by differential equations. Recently, INRs have also been introduced into PDE modeling and operator learning. CORAL (Serrano et al., 2023) represents functions on general geometries using coordinate-based neural fields and learns operators in the latent space of these representations, improving flexibility across irregular domains and samplings. GridMix (Wang et al., 2025) further studies spatial modulation for neural fields in PDE modeling by combining grid-based representations to preserve locality while modeling global structures. Different from these works, our method uses an INR-style SDF auto-decoder as a stable and controllable geometry representation, and further couples it with a geometry-informed field predictor to enable

differentiable shape optimization and inversion.

## 3. Method

**Problem setup.** We study PDE-governed systems whose computational domain $\Omega(\Gamma)$ is determined by an unknown geometry $\Gamma$. Given $\Gamma$, solving the PDE on $\Omega(\Gamma)$ produces physical fields $\mathbf{u}(\cdot;\Gamma)$. Our goal is to obtain an optimal geometry $\Gamma^\star$ by minimizing a task-dependent functional,

$$\Gamma^\star = \arg\min_\Gamma \ \mathcal{J}\big(\mathbf{u}(\cdot;\Gamma)\big), \tag{1}$$

where $\mathcal{J}$ denotes a performance objective for optimization or an observation-matching objective for inversion.

### 3.1. Overview of GANO

Solving Eq. 1 efficiently and reliably relies on three key components: (i) a *geometry representation* for $\Gamma$ that admits a compact latent space and stable updates, (ii) a *forward surrogate* to predict the physical fields $\mathbf{u}(\cdot;\Gamma)$, and (iii) an *optimization mechanism* that updates $\Gamma$ given task-specific objectives. To this end, we propose the **G**eometry-**A**ware **N**eural **O**ptimizer (**GANO**), a framework that enables *full cycle end-to-end differentiability* and *automation* for geometry optimization. GANO couples a compact implicit geometry representation (Sec. 3.2) with a geometry-informed field predictor (Sec. 3.3), and performs a remeshing-free optimization in the latent space (Sec. 3.4), which supports flexible, region-aware objectives.

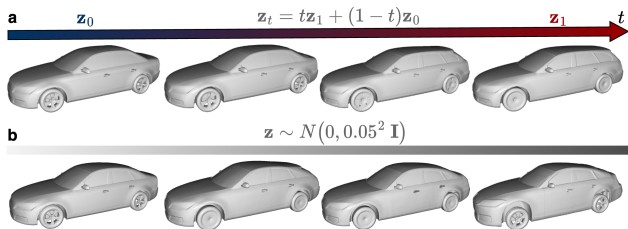

*Figure 3.* **Latent-space of STABLESDF. a**. Linear interpolation between two latent codes $\mathbf{z}_0$ and $\mathbf{z}_1$ in the dataset. **b**. Sampling latent codes from $\mathbf{z} \sim \mathcal{N}(\mathbf{0}, 0.05^2\mathbf{I})$.

## 3.2. Geometry Representation with STABLESDF

We build on DEEPSDF (Park et al., 2019) as a standard implicit representation, and introduce a simple but effective denoising technique to make the latent-to-geometry mapping locally smooth, which is critical for stable gradient-based optimization. We propose our adapted model as **STABLESDF**.

**Preliminary: DEEPSDF.** A *signed distance function* (SDF) assigns each spatial location $\mathbf{x} \in \mathbb{R}^3$ a signed distance to a surface $\Gamma$: $s(\mathbf{x}) = 0$ on $\Gamma$, $s(\mathbf{x}) < 0$ inside, and $s(\mathbf{x}) > 0$ outside, with $|s(\mathbf{x})| = \mathrm{dist}(\mathbf{x}, \Gamma)$ denoting the Euclidean distance to $\Gamma$. DEEPSDF represents a shape by learning a neural implicit decoder that predicts the signed distance given a coordinate $\mathbf{x}$ and a shape embedding $\mathbf{z}$:

$$s = s_\theta(\mathbf{x}, \mathbf{z}), \qquad \mathbf{x} \in \mathbb{R}^3,\ \mathbf{z} \in \mathbb{R}^d,\ s \in \mathbb{R}, \quad (2)$$

where $\mathbf{z}$ is the latent code corresponding to a specific geometry and $\theta$ denotes shared network parameters.

**STABLESDF.** To enable stable latent-space optimization and inversion, we propose an SDF-based auto-decoder with a denoising mechanism, which we term STABLESDF. Given a training set of geometries $\{\Gamma_i\}_{i=1}^N$, we link each geometry $\Gamma_i$ with a learnable latent code $\mathbf{z}_i \in \mathbb{R}^d$ and learn a shared decoder $s_\theta$. For each $\Gamma_i$, we sample points $\mathbf{x} \in \mathbb{R}^3$ in surrounding space and compute target signed distances $s$. We jointly optimize the decoder parameters and the latent codes using a $\ell_1$ reconstruction loss with a $\ell_2$ latent regularizer,

$$\min_{\theta, \{\mathbf{z}_i\}} \sum_{i=1}^N \mathbb{E}_{(\mathbf{x},s) \sim \mu_i} \left[ \left| s_\theta(\mathbf{x}, \tilde{\mathbf{z}}_i) - s \right| \right] + \lambda \|\mathbf{z}_i\|_2^2, \quad (3)$$

where $\mu_i$ denotes the sampling distribution associated with geometry $\Gamma_i$, and the $\ell_2$ term corresponds to a Gaussian prior on the latent codes.

A key design in STABLESDF is adding small Gaussian perturbations to the latent code during training, i.e., $\tilde{\mathbf{z}}_i = \mathbf{z}_i + \boldsymbol{\epsilon}$ with $\boldsymbol{\epsilon} \sim \mathcal{N}(\mathbf{0}, \sigma^2\mathbf{I})$. This denoising-style augmentation encourages the decoder to reconstruct valid geometry not only at the single point $\mathbf{z}_i$, but also throughout its local neighborhood. Consequently, the induced mapping from $\mathbf{z}$

to geometry $\Gamma(\mathbf{z}) = \{\mathbf{x} \mid s_\theta(\mathbf{x}, \mathbf{z}) = 0\}$ becomes smoother and less sensitive to perturbations, which can ensure the optimization loop in the geometry-consistent latent space. As shown in Fig. 3**a**, smoothly varying $\mathbf{z}$ leads to coherent and meaningful deformations of $\Gamma(\mathbf{z})$.

## 3.3. GI-TRANSOLVER for Forward Analysis

A central requirement of GANO is a *geometry-conditioned* forward surrogate that (i) accurately predicts solution fields on irregular discretizations and (ii) remains *differentiable* with respect to latent $\mathbf{z}$. We incorporate the TRANSOLVER (Wu et al., 2024a) backbone with an efficient geometry-informed mechanism, termed **GI-TRANSOLVER**, which not only explicitly injects geometry information but creates an effective gradient path from the optimization objectives to the latent code $\mathbf{z}$.

**Preliminary: TRANSOLVER.** Let the query set be $\mathcal{Q} = \{\mathbf{q}_i\}_{i=1}^N$, where each $\mathbf{q}_i$ contains spatial coordinates on an irregular mesh, optionally together with observed physical quantities. TRANSOLVER first embeds each query point into a feature vector $\mathbf{h}_i \in \mathbb{R}^D$. It introduces a Physics-Attention mechanism that adaptively groups points into $G$ learnable *slices*, where each slice is intended to represent an intrinsic physical state. Specifically, the slice assignment is produced by a point-wise projection followed by a Softmax over the slice dimension:

$$\boldsymbol{\ell}_i = \mathbf{W}_s\mathbf{h}_i,\ \mathbf{w}_i = \mathrm{Softmax}(\boldsymbol{\ell}_i),\ \sum_{g=1}^G w_{i,g} = 1, \quad (4)$$

where $\mathbf{W}_s$ is a learnable projection and $w_{i,g}$ denotes the degree to which point $\mathbf{q}_i$ is assigned to the $g$-th slice.

For each slice, TRANSOLVER then normalizes the assignment weights across points and aggregates point features into a physics-aware token:

$$\alpha_{i,g} = \frac{w_{i,g}}{\sum_{j=1}^N w_{j,g} + \varepsilon}, \quad (5)$$

$$\mathbf{t}_g = \sum_{i=1}^N \alpha_{i,g} \mathbf{W}_v\mathbf{h}_i, \qquad g = 1, \ldots, G, \quad (6)$$

where $\mathbf{W}_v$ is a learnable value projection.

The token set $\mathbf{T} = \{\mathbf{t}_g\}_{g=1}^G$ is then processed by standard self-attention in the compact slice-token space:

$$\tilde{\mathbf{T}} = \mathrm{Attention}(\mathbf{T}), \quad (7)$$

Finally, the updated tokens are broadcast back to point-level features through the original slice assignment weights:

$$\tilde{\mathbf{h}}_i = \sum_{g=1}^G w_{i,g} \mathbf{W}_o\tilde{\mathbf{t}}_g, \quad (8)$$

where $\mathbf{W}_o$ is an output projection. A point-wise prediction head then maps $\tilde{\mathbf{h}}_i$ to the target physical quantity $\hat{\mathbf{u}}(\mathbf{q}_i)$.

**GI-TRANSOLVER.** Geometry optimization requires the forward surrogate to be differentiable with respect to $\mathbf{z}$ so that the objective, depending on predicted fields, can update geometry through gradient descent. However, TRANSOLVER does not take *compact* geometry information as input, so gradients from model outputs cannot backpropagate to latent code $\mathbf{z}$. Therefore, we propose geometry-informed Transolver (**GI-TRANSOLVER**), which injects geometry information into the slice-token space through a gated mechanism, making the predicted fields explicitly differentiable with respect to $\mathbf{z}$ produced by STABLESDF.

Given a slice token $\mathbf{t}_g \in \mathbb{R}^D$ and latent code $\mathbf{z} \in \mathbb{R}^d$, we first project $\mathbf{z}$ to token dimension $D$, then compute a gate vector and use it to modulate the latent code, after which the modulated $\mathbf{z}$ is injected through a residual update:

$$\mathbf{g}_g = \sigma\left(\mathbf{W}_2 \operatorname{SiLU}\left(\mathbf{W}_1 \mathbf{t}_g\right)\right) \in (0,1)^D, \qquad (9)$$

$$\Delta \mathbf{t}_g = \mathbf{g}_g \odot \mathbf{W}_z \mathbf{z}, \quad \hat{\mathbf{t}}_g \leftarrow \mathbf{t}_g + \Delta \mathbf{t}_g. \qquad (10)$$

Importantly, the injection in Eqs. 9-10 is fully differentiable, enabling the information from optimization target to be passed to $\mathbf{z}$ efficiently, i.e., $\partial_{\mathbf{z}}\mathcal{J} = \frac{\partial\mathcal{J}}{\partial\hat{\mathbf{u}}}\frac{\partial\hat{\mathbf{u}}}{\partial\mathbf{z}}$.

## 3.4. Differentiable Optimization with GANO

We perform *gradient-based* geometry optimization or inversion in latent space by iteratively updating the latent code $\mathbf{z}$, freezing the geometry decoder $s_\theta$ and the forward surrogate. At each iteration, we maintain a surface point set $\mathcal{X}_t = \{\mathbf{x}_i\}_{i=1}^N \subset \Gamma(\mathbf{z}_t)$ and evaluate task objectives using predicted physical fields at these *boundary-consistent* query points, then backpropagating gradients to update $\mathbf{z}$. Notably, predicting *full physical fields* (instead of a fixed scalar) allows objectives $\mathcal{J}$ to be defined flexibly from field-level quantities (such as local properties). Moreover, this design enables (i) *part-wise control* via null-space projection, and (ii) *remeshing-free* iterations via SDF-based projection.

**Part-wise control via null-space projection.** In many real-world scenarios, only a subset of the geometry is allowed to change, while other components are expected to remain fixed. For example, drag minimization in vehicle aerodynamics often optimizes the body shape while keeping standardized parts (e.g., wheels and side mirrors) fixed. To achieve part-wise control, we specify a set of constraint points $\mathcal{X}_{\text{const}} = \{\mathbf{x}_m\}_{m=1}^M$ on components that should remain fixed. Owing to the generative ability of STABLESDF (Fig. 3b), by preserving these points during latent updates, we can keep the geometry of the corresponding parts. Let

$$\mathbf{G}(\mathbf{z}) = \frac{\partial s_\theta(\mathcal{X}_{\text{const}}, \mathbf{z})}{\partial \mathbf{z}} \in \mathbb{R}^{M \times d}. \qquad (11)$$

Given the unconstrained gradient $\mathbf{g} = \nabla_{\mathbf{z}}\mathcal{J}(\mathbf{z})$, we project it onto the null space of $\mathbf{G}(\mathbf{z})$:

$$\mathbf{P}(\mathbf{z}) = \mathbf{I} - \mathbf{G}(\mathbf{z})^\dagger \mathbf{G}(\mathbf{z}), \qquad \mathbf{g}_{\text{safe}} = \mathbf{P}(\mathbf{z})\,\mathbf{g}, \qquad (12)$$

where $\mathbf{G}^\dagger$ is the Moore–Penrose pseudo-inverse. This ensures $\mathbf{G}(\mathbf{z})\,\mathbf{g}_{\text{safe}} = \mathbf{0}$, suppressing first-order changes of the signed distances at $\mathcal{X}_{\text{const}}$ and thereby protecting the specified parts (detailed in Appendix A.4).

**Remeshing-free projection for boundary-consistent queries.** The surface $\Gamma(\mathbf{z})$ changes after updating $\mathbf{z}$, so the previous samples $\mathcal{X}_t$ drift off the new boundary. We maintain boundary consistency by adjusting sample points back to the updated $\Gamma(\mathbf{z})$ using an SDF-based projection to replace time-consuming remeshing:

$$\mathbf{x} \leftarrow \mathbf{x} - s_\theta(\mathbf{x}, \mathbf{z}) \frac{\nabla_{\mathbf{x}} s_\theta(\mathbf{x}, \mathbf{z})}{\|\nabla_{\mathbf{x}} s_\theta(\mathbf{x}, \mathbf{z})\|_2^2 + \varepsilon}. \qquad (13)$$

Eq. 13 is a first-order step that moves $\mathbf{x}$ toward nearest $s_\theta(\mathbf{x}, \mathbf{z}) = 0$ along the local SDF gradient direction (Appendix E.6). In practice, only a few projection steps are needed, yielding an efficient operator that keeps $\mathcal{X}_t$ boundary-consistent throughout iterations while avoiding explicit mesh reconstruction (detailed in Appendix A.5).

**Optimization loop.** Starting from the initial code $\mathbf{z}_0$ and samples $\mathcal{X}_0 \subset \Gamma(\mathbf{z}_0)$, each iteration (i) predicts fields on $\mathcal{X}_t$, (ii) calculates a task objective $\mathcal{J}(\mathbf{z}_t)$, (iii) back-propagates to obtain $\nabla_{\mathbf{z}}\mathcal{J}$ and optionally uses null-space projection for part-wise control, (iv) updates $\mathbf{z}_t$ to $\mathbf{z}_{t+1}$, and (v) projects $\mathcal{X}_t$ onto $\Gamma(\mathbf{z}_{t+1})$ using remeshing-free projection. Alg. 1 in Appendix C summarizes the procedure.

## 4. Theory

We provide a theoretical explanation for the effectiveness and stability of GANO. Our analysis begins with the denoising mechanism in STABLESDF, which induces a Jacobian regularization and thus reduces its sensitivity to latent perturbations. Under mild assumptions, this leads to smooth surface deformations across optimization iterations. Moreover, this sensitivity control bounds the bias introduced by the stopping-gradients operation during optimization. Formal theorems and proofs can be seen in Appendix A.

**(1) Denoising mechanism regularizes decoder sensitivity.** During STABLESDF training, we inject Gaussian noise into each latent code (Eq. 3). Theorem A.1 (Appendix A.1) shows that, in the small-noise regime and with mild local smoothness of $f(\mathbf{z}) \triangleq s_\theta(\mathbf{x}, \mathbf{z})$, we have

$$\mathbb{E}[|f(\mathbf{z}+\boldsymbol{\varepsilon}) - s|] \leq |f(\mathbf{z}) - s| + \sigma\sqrt{\frac{2}{\pi}}\|\nabla_{\mathbf{z}} f(\mathbf{z})\|_2 + \mathcal{O}(\sigma^2). \qquad (14)$$

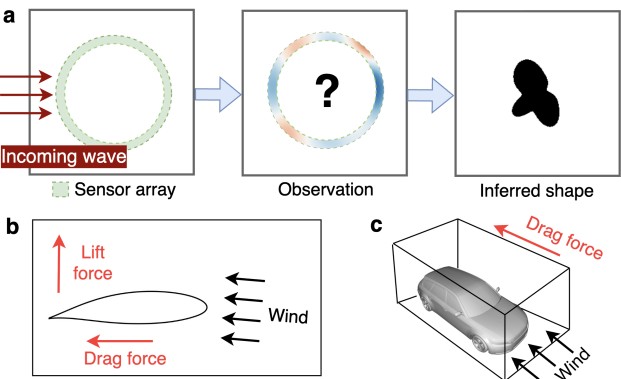

*Figure 4.* **Experimental setups. a**. *2D Helmholtz;* **b**. *2D airfoil;* **c**. *3D vehicle.*

*Table 1.* **Comparison of relative errors for forward analysis.** Best results are in **bold** and second-best results are underlined.

| Model | 2D Helmholtz | | 2D AirFoil | | 3D vehicle | |
|---|---|---|---|---|---|---|
| | Rel. L1 | Rel. L2 | Rel. L1 | Rel. L2 | Rel. L1 | Rel. L2 |
| TRANSOLVER | 0.0242 | 0.0267 | 0.0027 | 0.0060 | 0.1708 | 0.1841 |
| TRANSOLVER++ | 0.0317 | 0.0351 | 0.0078 | 0.0154 | 0.1799 | 0.1920 |
| AEROGTO | 0.0212 | 0.0234 | 0.0021 | 0.0057 | 0.1678 | 0.1790 |
| POINTNET++ | 0.0427 | 0.0462 | 0.0305 | 0.0409 | 0.3100 | 0.4004 |
| FNO | 0.0317 | 0.0299 | / | / | / | / |
| U-NET | 0.1884 | 0.1929 | 0.0150 | 0.0137 | / | / |
| DEEPONET | 0.1749 | 0.1656 | 0.0198 | 0.0340 | / | / |
| **GANO (ours)** | **0.0171** | **0.0170** | **0.0008** | **0.0022** | **0.1655** | **0.1782** |

Therefore, the additional penalty due to perturbation is approximately $\sigma\sqrt{\frac{2}{\pi}}\|\nabla_{\mathbf{z}}s_\theta(\mathbf{x},\mathbf{z})\|_2$, which acts as an implicit latent-Jacobian regularizer. This makes the decoder less sensitive to latent perturbations near the geometry manifold.

**(2) Reduced sensitivity yields controlled surface displacement.** Assume (i) level-set regularity $\|\nabla_{\mathbf{x}}s_\theta\| \geq m$ on $\Gamma(\mathbf{z})$ and (ii) a latent-sensitivity bound $\|\nabla_{\mathbf{z}}s_\theta\| \leq L_z$ in a tubular neighborhood of $\Gamma(\mathbf{z})$. Then, for sufficiently small $\delta\mathbf{z}$, the surface discrepancy $d_H(\cdot,\cdot)$ satisfies a first-order bound (Theorem A.4 in Appendix A.2):

$$d_H\big(\Gamma(\mathbf{z}),\Gamma(\mathbf{z}+\delta\mathbf{z})\big) \; \leq \; \frac{L_z}{m}\,\|\delta\mathbf{z}\|_2 + \mathcal{O}(\|\delta\mathbf{z}\|_2^2). \quad (15)$$

Therefore, reducing latent-sensitivity $L_z$ makes each latent update produce a more controlled geometric deformation.

**(3) Why detaching remeshing-free projection yields effective optimization.** Our optimization maintains boundary-consistent surface samples $X(\mathbf{z}) \subset \Gamma(\mathbf{z})$ via a projection operator in Eq. 13, while gradients are taken only through the explicit differentiable path $\mathbf{z} \rightarrow \hat{\mathbf{u}}$ created by GI-TRANSOLVER. Let $\mathcal{J}(\mathbf{z}) = \hat{\mathcal{J}}(\mathbf{z},X(\mathbf{z}))$; then the total derivative follows the chain rule:

$$\nabla_{\mathbf{z}}\mathcal{J}(\mathbf{z}) = \underbrace{\nabla_{\mathbf{z}}\hat{\mathcal{J}}(\mathbf{z},X)}_{\text{explicit path } \mathbf{z}\rightarrow\hat{\mathbf{u}}} + \underbrace{\left(\nabla_X\hat{\mathcal{J}}(\mathbf{z},X)\right)\frac{dX(\mathbf{z})}{d\mathbf{z}}}_{\text{geometry transport via projection}}. \quad (16)$$

In practice, we use the first term as our optimization gradient and stop gradients through the projection pathway, thus omitting the transport term. Theorem A.8 in Appendix A.3 shows that the norm of the omitted term is bounded by the same sensitivity that governs surface motion. Therefore, reducing latent sensitivity (smaller $L_z$) also reduces the bias introduced by detaching the projection of $X(\mathbf{z})$.

## 5. Experiments

### 5.1. Experimental Setup

**Benchmarks.** We evaluate **GANO** on three benchmarks that cover *forward prediction* and *geometry inversion/optimization* across 2D/3D geometries (Fig. 4). First, we consider a *2D Helmholtz inverse scattering* problem (Fig. 4**a**). The *forward task* is to predict the scattered field given the geometry, while the *inverse task* is to reconstruct the obstacle shape from boundary measurements. Second, we study *2D airfoil aerodynamics* on irregular meshes using AirFoil_9k (Ramos et al., 2023) (Fig. 4**b**) with the *forward task* to predict velocity and pressure fields, and the *optimization task* to update the airfoil geometry to maximize lift with a soft drag constraint. Third, we conduct large-scale *3D vehicle* experiments on DrivAerNet++ (Elrefaie et al., 2024) (Fig. 4**c**). The *forward task* predicts surface pressure, and the *optimization task* minimizes drag. More details can be seen in Appendix D. **The code will be released at `https://github.com/intell-sci-comput/GANO`.**

**Baselines.** We compare **GANO** against representative neural surrogates and differentiable geometry-optimization pipelines. For forward prediction, we include Transformer-based surrogates **TRANSOLVER** (Wu et al., 2024a) and **TRANSOLVER++** (Luo et al., 2025), the generic point-set encoder **POINTNET++** (Qi et al., 2017), and **AEROGTO** (Liu et al., 2025) as a graph-transformer operator. For 2D inversion and optimization, we further compare to three surrogate-based optimization baselines built on **DEEPONET** (Shukla et al., 2024), **U-Net** (Rehmann et al., 2025), and **FNO** (Li et al., 2021), which predict full fields from geometric inputs and optimize objectives via back-propagation. For further comparison, we also conduct geometry code injection to **DEEPONET**, **U-Net**, and add **CORAL** as representative INR method on 2D Helmholtz task. Details could be seen in Appendix F. For 3D optimization, we compare our method with **PHYSGEN** including pressure prediction and optimization results.

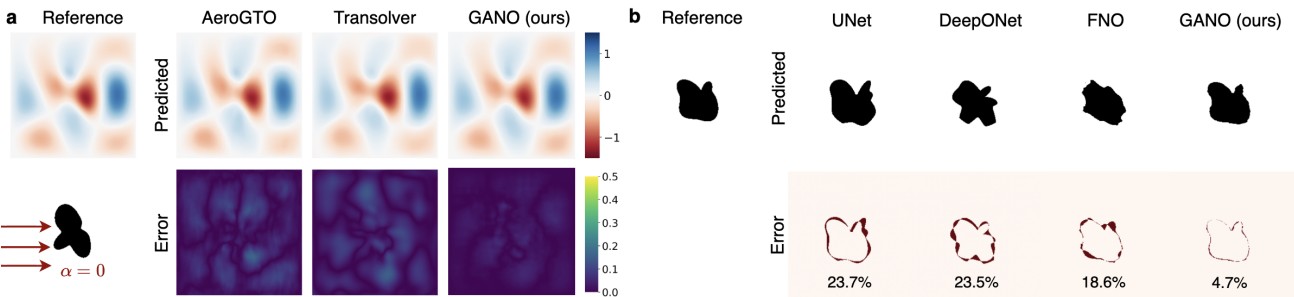

*Figure 5.* **2D Helmholtz: forward prediction and shape inversion. a**. Comparison of the real-part. **b**. Shape inversion from sensor array.

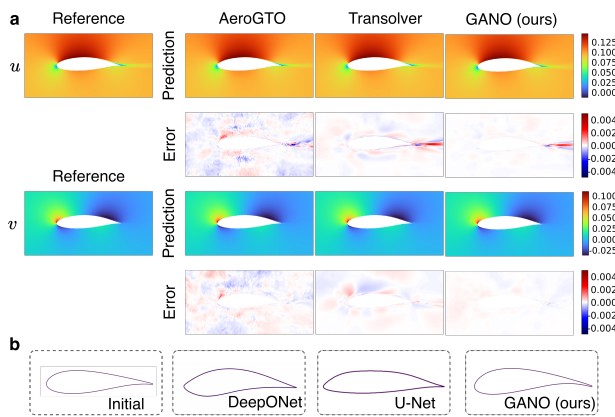

*Figure 6.* **2D Airfoil: forward prediction and shape optimization. a**. Comparison of predicted flow fields. **b**. Optimized airfoil shapes under soft drag constraint ($C_D < 0.02$). The shown airfoils are the best results from 100 iterations of each method.

*Table 2.* **Optimization results on 2D airfoil.** The soft drag constraint is $C_D < 0.02$.

|  | Initial | DEEPONET | U-NET | GANO (ours) |
|---|---|---|---|---|
| $C_L$ (↑) | 0.75 | 0.93 | 1.20 | 1.42 |
| $C_D$ ($< 0.02$) | 0.014 | 0.021 | 0.016 | 0.017 |
| $C_L/C_D$ (↑) | 53.6 (+0.0%) | 44.3 (-17.3%) | 75.0 (+40.0%) | 83.4 (+55.9%) |

### 5.2. 2D Helmholtz Equation

We benchmark *forward scattering prediction* on the 2D Helmholtz benchmark. **GANO** achieves the lowest errors among all baselines, outperforming the strongest model AEROGTO and TRANSOLVER/TRANSOLVER++ (Table 1). Fig. 5**a** shows qualitative comparisons, where **GANO** yields noticeably reduced error maps. In the *inverse problem*, the objective is to recover the unknown obstacle from sparse observations on a circle (Fig. 4**a**). Compared with differentiable inversion baselines (Fig. 5**b**), **GANO** produces shapes with finer-scale details and exhibits substantially smaller reconstruction errors. We fix the sensor array size at 100 in this case, and ablations with different sensor numbers can be seen in Appendix E.3.

### 5.3. 2D Airfoil Aerodynamics

In this section, we assess *forward aerodynamic prediction* on AirFoil_9k. **GANO** achieves the best accuracy of Rel.L2 0.0022 much lower than the second best AEROGTO of 0.0057. We further conduct airfoil optimization by maximizing lift under a penalty-based soft drag constraint ($C_D < 0.02$). As shown in Fig. 6, our model exhibits more aerodynamically reasonable shape characteristics while preserving the smoother leading edge of the original design, which is favorable in our setting. Quantitatively, Table 2 shows that **GANO** achieves the highest lift-to-drag ratio $C_L/C_D = 83.4$, surpassing U-Net (75.0) and DeepONet (44.3). Optimization results are validated by COMSOL.

### 5.4. 3D Vehicle Aerodynamics on DrivAerNet++.

To evaluate under more complex 3D conditions, we use DrivAerNet++ as a challenging benchmark. Table 1 shows that **GANO** achieves the best performance. Fig. 7**a** provides a qualitative comparison. These results indicate that geometric injection remains effective at large scale and under the challenging, irregular dense point clouds. We further validate stable latent-space vehicle optimization by reducing a drag proxy in line with (Hao et al., 2025; You et al., 2025) while constraining the geometry to avoid deviating excessively from the initial design (optionally with local protection via null-space projection; Appendix A.4). We provide detailed visual comparisons for two vehicle categories (estateback and fastback) in Fig. 7**b-c**. The optimized shapes are plausible while maintaining industry-level geometric fidelity, preserving key design elements. High-fidelity Open-FOAM simulations verify that our method reduces the drag coefficient $C_D$ from 0.346 to 0.323 (6.64% ↓) for the estateback and from 0.299 to 0.278 (7.02% ↓) for the fastback, respectively. We use PHYSGEN to optimize the same cars and Fig. 8 demonstrates that GANO maintains better drag reduction and practicability. Typically, PhysGen often reduces drag by shrinking or sweeping back the side mirrors, while GANO largely preserves them due to null-space projection while still achieving meaningful drag reduction. The enlarged views highlight this difference.

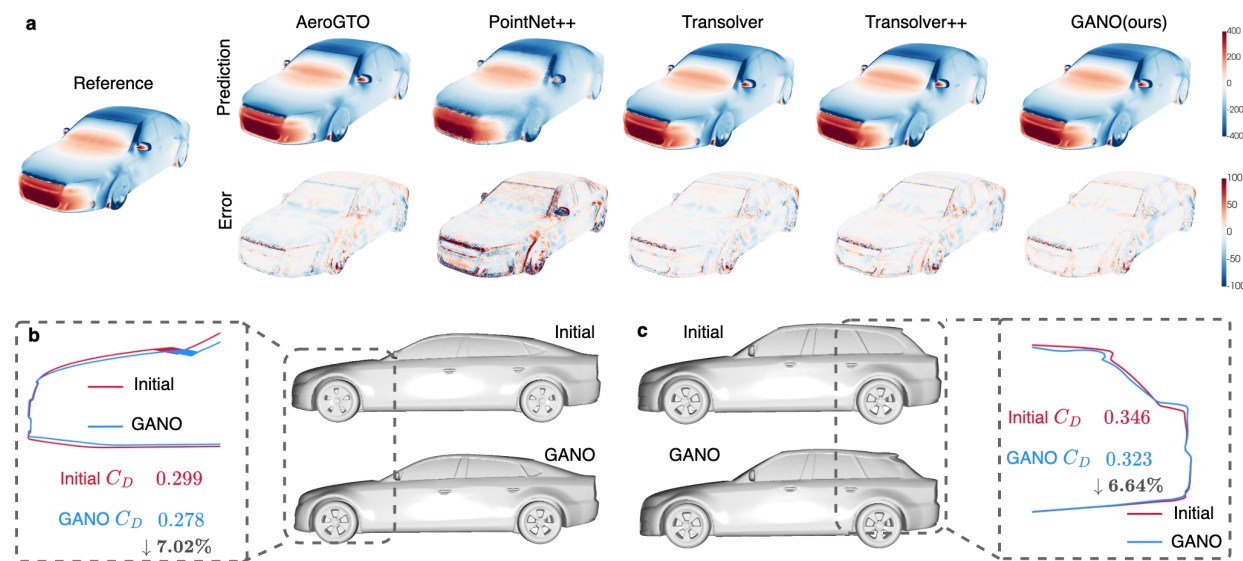

*Figure 7.* **3D Vehicle: surface pressure prediction and optimization on DrivAerNet++. a.** Comparison of surface pressure. **b-c.** Two representative optimization cases. For each case, we show the initial shape and the optimized result from GANO.

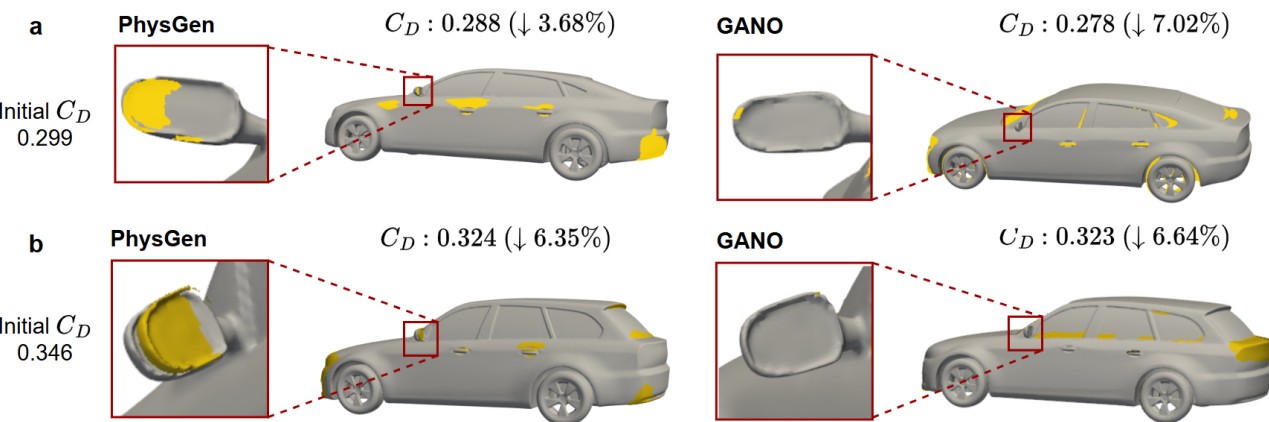

*Figure 8.* **Comparison of GANO and PhysGen on vehicle optimization.** Gray denotes the input vehicle, and yellow highlights regions with large geometric changes after optimization. **a.** GANO achieves a better result than PhysGen; **b.** The two are comparable. However, PhysGen often reduces drag by shrinking or sweeping back the side mirrors, while GANO largely preserves them due to null-space projection while still achieving meaningful drag reduction. The enlarged views highlight this difference.

## 6. Model Analysis

### 6.1. Latent Space Properties

We propose STABLESDF to obtain a continuous and compact representation. As shown in Theorem A.1, injecting Gaussian noise during training approximates a penalty on the decoder's latent Jacobian norm $\|\nabla_{\mathbf{z}} s_\theta(\mathbf{x}, \mathbf{z})\|$, thereby improving robustness to latent perturbations. Consistent with this analysis, Fig. 9**a** shows that the empirical distribution of $\|\nabla_{\mathbf{z}} s_\theta(\mathbf{x}, \mathbf{z})\|$ for STABLESDF concentrates at substantially smaller values than that of standard DEEPSDF. We further probe robustness by perturbing a latent code $\mathbf{z}_0$ from a test vehicle with Gaussian noise (Fig. 9**b**). At a moderate noise level ($5 \times 10^{-2}$), DEEPSDF exhibits severe

deformation of fine details such as mirrors and door handles, whereas STABLESDF largely preserves these structures and maintains a coherent surface. Furthermore, we conduct ablation studies on the level of noise in Appendix E.2. And Appendix F compares results after optimizing the same Fastback and Estateback car using StableSDF and DeepSDF, respectively, showing StableSDF's advantage during optimization.

### 6.2. Necessity of Auto-Decoder Formulation

To assess whether an auto-decoder architecture is necessary for high-quality geometric modeling, we train a variational autoencoder (VAE) baseline (Kingma & Welling,

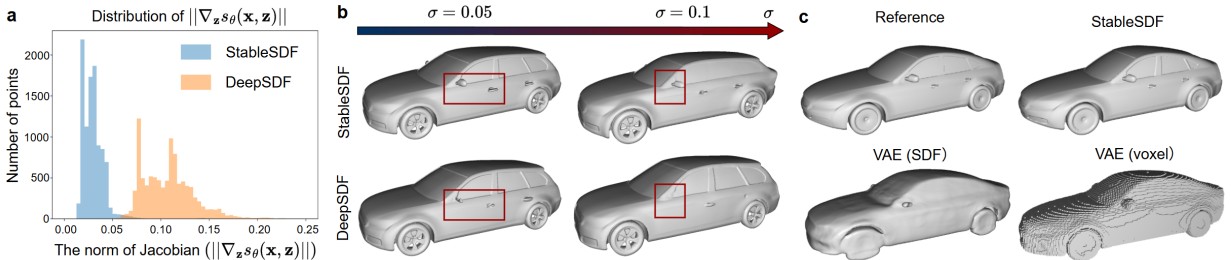

*Figure 9.* **Analysis of STABLESDF. a**. The distribution of latent Jacobian norms $\|\nabla_{\mathbf{z}} s_\theta(\mathbf{x}, \mathbf{z})\|$ for DEEPSDF and STABLESDF. **b**. Decoded shapes under increasing Gaussian perturbations around a test latent code. **c**. Reconstruction comparison on a test vehicle.

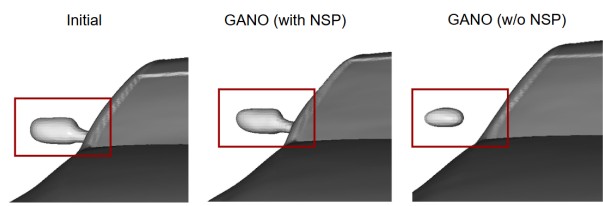

*Figure 10.* **Null-space projection (NSP) preserves the mirror during optimization.** Mirror close-up for the initial design and GANO optimized with/without NSP.

2014) that uses a tri-plane representation to parameterize the latent space. The VAE takes surface point clouds as input and is trained with two output targets: (i) SDF and (ii) occupancy (voxel). We find that SDF prediction with the VAE is difficult to optimize and often collapses into high-deformation reconstructions, whereas occupancy prediction is more stable. However, occupancy is a binary voxel signal and discards fine-grained geometric details. Visual comparison could be found in Fig. 9**c**. These results support our choice of an auto-decoder SDF formulation as a more suitable approach for detailed geometry reconstruction.

### 6.3. The Effects of Null-space Projection (NSP)

In industrial optimization, preserving specific sub-components is often critical, motivating the design of our NSP technique in GANO. To validate the effectiveness of NSP, we optimize the same vehicle under two settings: (i) **with NSP**, where we sample constraint points on the mirrors and project each update of $\mathbf{z}$ into the null space of the corresponding constraint Jacobian; (ii) **without NSP**, where we apply the same updates without geometric constraints. Fig.10 shows that after optimization, the baseline (w/o NSP) exhibits severe mirror deformation and detachment from the car body, while the projected variant preserves the mirror geometry. These results demonstrate that NSP effectively protects local geometric details during optimization.

### 6.4. Robustness to the Number of Input Points

We further evaluate robustness under varying point-cloud densities. All models are trained on $5 \times 10^4$ points and

*Table 3.* Error metrics under different numbers of points.

| Model | Number of points ($\times 10^3$) | | | | | | |
|---|---|---|---|---|---|---|---|
| | 1 | 5 | 10 | 50 | 100 | 200 | 500 |
| TRANSOLVER | 27.5 | 19.3 | 18.6 | 18.4 | 17.6 | 17.6 | 17.6 |
| AEROGTO | 64.0 | 38.7 | 26.6 | 17.9 | 18.1 | 20.3 | 26.9 |
| GANO (ours) | **20.4** | **18.1** | **17.9** | **17.8** | **17.2** | **17.2** | **17.2** |

tested on point clouds ranging from sparse ($10^3$ points) to very dense ($5 \times 10^5$ points). GI-TRANSOLVER remains stable across this range, benefiting from explicit geometric injection, while TRANSOLVER degrades notably under sparse inputs. AEROGTO performs poorly in both sparse and dense regimes, indicating weaker invariance to query density. Since high-fidelity CFD typically requires dense meshes, our results are particularly encouraging: a model trained with relatively modest resolution can generalize to substantially denser point sets, thereby reducing data and computational costs during training.

### 6.5. Additional Experiments

Appendix E details ablation studies for GI-TRANSOLVER and STABLESDF, and the analysis of remeshing-free projection as well as computational cost.

## 7. Conclusion

We presented **GANO**, an end-to-end differentiable framework that unifies geometry representation, field-level prediction, and automated shape optimization/inversion in a single latent-space loop. By combining a STABLESDF-style auto-decoder and a geometry-informed surrogate that provides a reliable gradient pathway, GANO enables efficient, remeshing-free geometry updates with part-wise control via null-space projection. GANO achieves state-of-the-art forward accuracy and delivers stable, controllable shape updates for both optimization and inversion. A current limitation is that GANO focuses on steady-state PDE settings and does not apply to spatiotemporal dynamics. A promising future work is extending GANO to *time-dependent* PDE systems and *dynamic* shape optimization.

## Acknowledgment

The work is supported by the National Natural Science Foundation of China (No. 62506367 and No. 62276269) and the Beijing Natural Science Foundation (No. F261002). R.Z. would like to acknowledge the supported from the China Postdoctoral Science Foundation under Grant Number 2025M771582 and the Postdoctoral Fellowship Program of CPSF under Grant Number GZB20250408.

## Impact Statement

This paper aims to advance machine learning methods for differentiable geometry modeling and physics-guided optimization. Potential societal impacts are similar to those of related work in surrogate modeling and design optimization. We do not anticipate immediate negative impacts that warrant specific discussion beyond standard considerations of responsible use.

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

## A. Theory Results

We provide detailed derivations and proofs for the theory presented in the main text here.

### A.1. Small-noise limit: latent perturbation training as latent-sensitivity regularization

**Overview.** This section analyzes the denoising mechanism used in STABLESDF, where isotropic Gaussian noise is injected into the latent code and an $\ell_1$ reconstruction loss is minimized (Eq. (3)). Although the pointwise $\ell_1$ loss is non-smooth, the Gaussian expectation yields a *Gaussian-smoothed* objective in latent space. We show that, in the small-noise regime, this perturbed objective can be well-approximated by a first-order linearization of the decoder output, with an explicit $\mathcal{O}(\sigma^2)$ remainder controlled by the local curvature. As a consequence, when reconstruction is accurate, latent perturbation training induces an implicit penalty proportional to $\|\nabla_{\mathbf{z}} s_\theta(\mathbf{x}, \mathbf{z})\|_2$, discouraging excessive decoder sensitivity to latent perturbations.

**Setup.** Fix a query input $\mathbf{x} \in \mathbb{R}^3$ and a target scalar $s \in \mathbb{R}$. Define the scalar decoder output

$$f(\mathbf{z}) \triangleq s_\theta(\mathbf{x}, \mathbf{z}), \qquad \mathbf{z} \in \mathbb{R}^d,$$

and the $\ell_1$ reconstruction loss

$$\ell(\mathbf{z}) \triangleq \big|f(\mathbf{z}) - s\big|.$$

Latent perturbation training injects isotropic Gaussian noise:

$$\tilde{\ell}_\sigma(\mathbf{z}) \triangleq \mathbb{E}_{\boldsymbol{\varepsilon} \sim \mathcal{N}(\mathbf{0}, \sigma^2 \mathbf{I}_d)}\big[\ell(\mathbf{z} + \boldsymbol{\varepsilon})\big] = \mathbb{E}\Big[\big|f(\mathbf{z} + \boldsymbol{\varepsilon}) - s\big|\Big].$$

**Main result.**

**Theorem A.1** (**Latent perturbation under an $\ell_1$ loss induces a latent-Jacobian-norm regularization**). *Assume $f$ is twice continuously differentiable and $\forall \mathbf{z}$ and $\xi$ its Hessian is uniformly bounded there:*

$$\big\|\nabla_{\mathbf{z}}^2 f(\boldsymbol{\xi})\big\|_{\mathrm{op}} \le M < \infty.$$

*where*

$$\|A\|_{\mathrm{op}} \triangleq \sup_{\|x\|_2 = 1} \|Ax\|_2 = \sup_{x \ne 0} \frac{\|Ax\|_2}{\|x\|_2}.$$

*Let $\boldsymbol{\varepsilon} \sim \mathcal{N}(\mathbf{0}, \sigma^2 \mathbf{I}_d)$ and denote $r \triangleq f(\mathbf{z}) - s$ and $g \triangleq \nabla_{\mathbf{z}} f(\mathbf{z})$. Then the perturbed $\ell_1$ loss admits the approximation bound*

$$\Big|\tilde{\ell}_\sigma(\mathbf{z}) - \mathbb{E}\big[\,|r + g^\top \boldsymbol{\varepsilon}|\,\big]\Big| \le \frac{M}{2}\mathbb{E}\big[\|\boldsymbol{\varepsilon}\|_2^2\big] = \frac{M}{2}\sigma^2 d. \tag{17}$$

*Moreover, since $g^\top \boldsymbol{\varepsilon} \sim \mathcal{N}(0, \sigma^2 \|g\|_2^2)$, we have*

$$\Big|\mathbb{E}\big[\,|r + g^\top \boldsymbol{\varepsilon}|\,\big] - \mathbb{E}\big[\,|g^\top \boldsymbol{\varepsilon}|\,\big]\Big| \le |r|, \qquad \mathbb{E}\big[\,|g^\top \boldsymbol{\varepsilon}|\,\big] = \sigma\sqrt{\frac{2}{\pi}}\,\|g\|_2. \tag{18}$$

*Proof.* By Taylor's theorem applied to $f$ at $\mathbf{z}$, for each $\boldsymbol{\varepsilon}$ there exists $t \in (0, 1)$ such that

$$f(\mathbf{z} + \boldsymbol{\varepsilon}) = f(\mathbf{z}) + g^\top \boldsymbol{\varepsilon} + \frac{1}{2}\boldsymbol{\varepsilon}^\top \nabla_{\mathbf{z}}^2 f(\mathbf{z} + t\boldsymbol{\varepsilon})\boldsymbol{\varepsilon}.$$

By the Hessian bound, we have

$$\Big|f(\mathbf{z} + \boldsymbol{\varepsilon}) - f(\mathbf{z}) - g^\top \boldsymbol{\varepsilon}\Big| \le \frac{1}{2}\big\|\nabla_{\mathbf{z}}^2 f(\mathbf{z} + t\boldsymbol{\varepsilon})\big\|_{\mathrm{op}}\|\boldsymbol{\varepsilon}\|_2^2 \le \frac{M}{2}\|\boldsymbol{\varepsilon}\|_2^2.$$

Let $a = f(\mathbf{z} + \boldsymbol{\varepsilon}) - s$ and $b = r + g^\top \boldsymbol{\varepsilon}$. Since the absolute value is 1-Lipschitz, $||a| - |b|| \le |a - b|$, hence

$$\Big||f(\mathbf{z} + \boldsymbol{\varepsilon}) - s| - |r + g^\top \boldsymbol{\varepsilon}|\Big| \le \frac{M}{2}\|\boldsymbol{\varepsilon}\|_2^2.$$

Taking expectation over $\boldsymbol{\varepsilon}$ yields Eq. (17). Finally, Eq. (18) follows from the triangle inequality $||r + t| - |t|| \le |r|$ and the identity $\mathbb{E}|\mathcal{N}(0, \tau^2)| = \tau\sqrt{2/\pi}$ with $\tau = \sigma\|g\|_2$. $\square$

**Corollary A.2** (**Near-optimal reconstruction: dominant latent-Jacobian-norm penalty**). *Under the assumptions of Theorem A.1, if $|f(\mathbf{z}) - s| \leq \eta$, then*

$$\left| \tilde{\ell}_\sigma(\mathbf{z}) - \sigma\sqrt{\frac{2}{\pi}} \left\| \nabla_{\mathbf{z}} f(\mathbf{z}) \right\|_2 \right| \; \leq \; \eta + \frac{M}{2}\sigma^2 d. \tag{19}$$

*Thus, for well-reconstructed samples (small $\eta$) and small noise $\sigma$, latent perturbation training effectively penalizes $\|\nabla_{\mathbf{z}} s_\theta(\mathbf{x}, \mathbf{z})\|_2$, discouraging excessive decoder sensitivity to latent changes.*

*Proof.* By Eq. (17) and Eq. (18),

$$\left| \tilde{\ell}_\sigma(\mathbf{z}) - \sigma\sqrt{\frac{2}{\pi}} \|g\|_2 \right| \leq \left| \tilde{\ell}_\sigma(\mathbf{z}) - \mathbb{E}|r + g^\top \varepsilon| \right| + \left| \mathbb{E}|r + g^\top \varepsilon| - \mathbb{E}|g^\top \varepsilon| \right| \leq \frac{M}{2}\sigma^2 d + |r| \leq \frac{M}{2}\sigma^2 d + \eta.$$

$\square$

**Remark.** If one replaces the $\ell_1$ loss by a smooth squared loss (or a smooth robust loss such as pseudo-Huber), a classical second-order small-noise expansion yields a quadratic latent-Jacobian penalty of the form $\sigma^2 \|\nabla_{\mathbf{z}} f(\mathbf{z})\|_2^2$. Our main text focuses on the $\ell_1$ setting used in STABLESDF.

## A.2. Controlled implicit-surface displacement under latent updates

**Overview.** This section shows that under a regular level-set condition and a bounded latent-sensitivity assumption for the implicit field $s_\theta(\mathbf{x}, \mathbf{z})$, a small latent update $\delta\mathbf{z}$ induces only a small displacement of the zero level set $\Gamma(\mathbf{z}) = \{\mathbf{x} : s_\theta(\mathbf{x}, \mathbf{z}) = 0\}$ in geometry. Measured by the Hausdorff distance, the surface displacement admits a first-order Lipschitz upper bound, namely $d_H(\Gamma(\mathbf{z}), \Gamma(\mathbf{z} + \delta\mathbf{z})) \lesssim (L_{\mathbf{z}}/m)\|\delta\mathbf{z}\|_2$, together with a second-order remainder term.

**Setup.** Let $s_\theta : \mathbb{R}^3 \times \mathbb{R}^d \to \mathbb{R}$ be a scalar implicit field (e.g., our STABLESDF model). Define the zero level set

$$\Gamma(\mathbf{z}) \triangleq \{\mathbf{x} \in \mathbb{R}^3 : \; s_\theta(\mathbf{x}, \mathbf{z}) = 0\}.$$

The Hausdorff distance between sets $A, B \subset \mathbb{R}^3$ is

$$d_H(A, B) \triangleq \max\left\{ \sup_{\mathbf{a} \in A} \inf_{\mathbf{b} \in B} \|\mathbf{a} - \mathbf{b}\|_2, \; \sup_{\mathbf{b} \in B} \inf_{\mathbf{a} \in A} \|\mathbf{a} - \mathbf{b}\|_2 \right\}.$$

**Regularity and bounded sensitivity.** Let $\mathcal{Z} \subset \mathbb{R}^d$ be a set of latent codes. For a set $S \subset \mathbb{R}^3$ and $\rho > 0$, define the tubular neighborhood $N_\rho(S) \triangleq \{\mathbf{x} : \text{dist}(\mathbf{x}, S) \leq \rho\}$.

**Assumption A.3** (**Regular level set and bounded latent sensitivity**). Assume $s_\theta$ is $\mathcal{C}^2$ in $(\mathbf{x}, \mathbf{z})$ on a region containing $\bigcup_{\mathbf{z} \in \mathcal{Z}} N_\rho(\Gamma(\mathbf{z})) \times B_{\rho_{\mathbf{z}}}(\mathbf{z})$ for some $\rho, \rho_{\mathbf{z}} > 0$. There exist constants $m > 0$ and $L_{\mathbf{z}} > 0$ such that for all $\mathbf{z} \in \mathcal{Z}$:

1. For all $\mathbf{x} \in \Gamma(\mathbf{z})$, $\|\nabla_{\mathbf{x}} s_\theta(\mathbf{x}, \mathbf{z})\|_2 \geq m$.

2. For all $\mathbf{x} \in N_\rho(\Gamma(\mathbf{z}))$, $\|\nabla_{\mathbf{z}} s_\theta(\mathbf{x}, \mathbf{z})\|_2 \leq L_{\mathbf{z}}$.

Moreover, there exist $M_{\mathbf{xz}}, M_{\mathbf{xx}} < \infty$ such that on the same region

$$\|\nabla_{\mathbf{xz}}^2 s_\theta(\mathbf{x}, \mathbf{z})\|_{\text{op}} \leq M_{\mathbf{xz}}, \qquad \|\nabla_{\mathbf{xx}}^2 s_\theta(\mathbf{x}, \mathbf{z})\|_{\text{op}} \leq M_{\mathbf{xx}}.$$

In addition, assume each $\Gamma(\mathbf{z})$ is contained in a common compact set $K \subset \mathbb{R}^3$ (e.g., due to a fixed bounding box of interest), so that $d_H(\Gamma(\mathbf{z}), \Gamma(\mathbf{z}')) < \infty$ is well-defined. Finally, to ensure uniformity over $\mathbf{z} \in \mathcal{Z}$, assume $\sup_{\mathbf{z} \in \mathcal{Z}} L_{\mathbf{z}} < \infty$.

**Main theorem.**

**Theorem A.4** (Latent-to-surface Lipschitz displacement bound). *Under Assumption A.3, there exists $\delta_0 > 0$ such that for all $\mathbf{z} \in \mathcal{Z}$ and $\delta\mathbf{z}$ with $\|\delta\mathbf{z}\|_2 \leq \delta_0$,*

$$d_H\big(\Gamma(\mathbf{z}), \Gamma(\mathbf{z} + \delta\mathbf{z})\big) \leq \frac{L_{\mathbf{z}}}{m}\|\delta\mathbf{z}\|_2 + \mathcal{O}\big(\|\delta\mathbf{z}\|_2^2\big), \tag{20}$$

*where the $\mathcal{O}(\|\delta\mathbf{z}\|_2^2)$ term is uniform over $\mathbf{z} \in \mathcal{Z}$.*

*Proof.* Fix $\mathbf{z} \in \mathcal{Z}$ and $\delta\mathbf{z}$ small. Pick any $\mathbf{x} \in \Gamma(\mathbf{z})$ so $s_\theta(\mathbf{x}, \mathbf{z}) = 0$ and define $r \triangleq s_\theta(\mathbf{x}, \mathbf{z} + \delta\mathbf{z})$. By the mean value theorem in $\mathbf{z}$ and Assumption A.3(2),

$$|r| = \big|\nabla_{\mathbf{z}} s_\theta(\mathbf{x}, \mathbf{z} + t\delta\mathbf{z})^\top \delta\mathbf{z}\big| \leq L_{\mathbf{z}}\|\delta\mathbf{z}\|_2 \quad \text{for some } t \in (0, 1). \tag{21}$$

Let the unit normal at $(\mathbf{x}, \mathbf{z})$ be

$$\mathbf{n} \triangleq \frac{\nabla_{\mathbf{x}} s_\theta(\mathbf{x}, \mathbf{z})}{\|\nabla_{\mathbf{x}} s_\theta(\mathbf{x}, \mathbf{z})\|_2}.$$

Consider $\phi(\alpha) \triangleq s_\theta(\mathbf{x} - \alpha\mathbf{n}, \mathbf{z} + \delta\mathbf{z})$. Then $\phi(0) = r$ and $\phi'(0) = -\nabla_{\mathbf{x}} s_\theta(\mathbf{x}, \mathbf{z} + \delta\mathbf{z})^\top \mathbf{n}$. Using Assumption A.3 and the mixed Hessian bound,

$$\nabla_{\mathbf{x}} s_\theta(\mathbf{x}, \mathbf{z} + \delta\mathbf{z})^\top \mathbf{n} \geq \|\nabla_{\mathbf{x}} s_\theta(\mathbf{x}, \mathbf{z})\|_2 - M_{\mathbf{xz}}\|\delta\mathbf{z}\|_2 \geq m - M_{\mathbf{xz}}\|\delta\mathbf{z}\|_2.$$

For $\|\delta\mathbf{z}\|_2$ sufficiently small, we have $\nabla_{\mathbf{x}} s_\theta(\mathbf{x}, \mathbf{z} + \delta\mathbf{z})^\top \mathbf{n} \geq m/2$. Define the Newton-style step

$$\alpha_0 \triangleq \frac{\phi(0)}{\nabla_{\mathbf{x}} s_\theta(\mathbf{x}, \mathbf{z} + \delta\mathbf{z})^\top \mathbf{n}} = \frac{r}{\nabla_{\mathbf{x}} s_\theta(\mathbf{x}, \mathbf{z} + \delta\mathbf{z})^\top \mathbf{n}}.$$

Combining with Eq. 21 yields

$$|\alpha_0| \leq \frac{L_{\mathbf{z}}}{m}\|\delta\mathbf{z}\|_2 + \mathcal{O}(\|\delta\mathbf{z}\|_2^2). \tag{22}$$

We now justify existence of a true root $\alpha^\star$ near $\alpha_0$. Since $s_\theta$ is $\mathcal{C}^2$ and $\|\nabla_{\mathbf{x}} s_\theta\| \geq m/2$ in a neighborhood for small $\|\delta\mathbf{z}\|_2$, the one-dimensional function $\phi$ is $\mathcal{C}^2$ with $\phi'(0) \leq -m/2$ and $|\phi''(\alpha)| \leq M_{\mathbf{xx}}$ for $\alpha$ in a sufficiently small interval (by the spatial Hessian bound). A quantitative implicit function / Newton–Kantorovich argument then guarantees that there exists a unique $\alpha^\star$ in a neighborhood of 0 such that $\phi(\alpha^\star) = 0$, and moreover $|\alpha^\star - \alpha_0| = \mathcal{O}(\alpha_0^2)$ as $\alpha_0 \to 0$. Consequently,

$$|\alpha^\star| \leq |\alpha_0| + |\alpha^\star - \alpha_0| \leq \frac{L_{\mathbf{z}}}{m}\|\delta\mathbf{z}\|_2 + \mathcal{O}(\|\delta\mathbf{z}\|_2^2).$$

Let $\mathbf{x}' \triangleq \mathbf{x} - \alpha^\star\mathbf{n} \in \Gamma(\mathbf{z} + \delta\mathbf{z})$. Then $\|\mathbf{x}' - \mathbf{x}\|_2 = |\alpha^\star|$ satisfies the same bound, implying

$$\sup_{\mathbf{x} \in \Gamma(\mathbf{z})} \inf_{\mathbf{y} \in \Gamma(\mathbf{z}+\delta\mathbf{z})} \|\mathbf{x} - \mathbf{y}\|_2 \leq \frac{L_{\mathbf{z}}}{m}\|\delta\mathbf{z}\|_2 + \mathcal{O}(\|\delta\mathbf{z}\|_2^2).$$

Repeating the argument with $(\mathbf{z} + \delta\mathbf{z})$ and $\mathbf{z}$ swapped gives the reverse directed bound. Taking the maximum of the two directed bounds yields Eq. 20. Uniformity over $\mathbf{z} \in \mathcal{Z}$ follows from the uniform constants in Assumption A.3 (including the compact containment ensuring finiteness of $d_H$). $\qquad\square$

**Corollary A.5** (Per-iteration geometric change under gradient descent). *Under Theorem A.4, for an update $\mathbf{z}_{t+1} = \mathbf{z}_t - \eta\mathbf{g}_t$ with $\eta\|\mathbf{g}_t\|_2 \leq \delta_0$,*

$$d_H\big(\Gamma(\mathbf{z}_t), \Gamma(\mathbf{z}_{t+1})\big) \leq \frac{L_{\mathbf{z}}}{m}\eta\|\mathbf{g}_t\|_2 + \mathcal{O}\big(\eta^2\|\mathbf{g}_t\|_2^2\big).$$

## A.3. Detached geometry updates during optimization

**Overview.** This section shows that when a downstream objective depends on both the latent code $\mathbf{z}$ and geometry-dependent samples $\mathbf{X}(\mathbf{z})$ induced by $\mathbf{z}$, the *full* reduced gradient decomposes into (i) a partial-gradient term that holds $\mathbf{X}$ fixed and (ii) a transport term that propagates through the geometry-feasibility map. Moreover, when the implicit-surface sensitivity is controlled (under the assumptions of the previous subsection), the transport term discarded by stop-gradient/detach through $\mathbf{X}(\mathbf{z})$ can be upper bounded, explaining why detached optimization provides a principled and stable approximation.

**Setup.** Let $\mathbf{X} = (\mathbf{x}_1, \ldots, \mathbf{x}_N) \in (\mathbb{R}^3)^N$ denote quadrature/sample points constrained to lie on the surface $\Gamma(\mathbf{z})^N$. Consider a generic downstream objective

$$\widehat{\mathcal{J}}(\mathbf{z}, \mathbf{X}) \triangleq \frac{1}{N} \sum_{i=1}^{N} \psi(\widehat{u}_\phi(\mathbf{x}_i, \mathbf{z}), \mathbf{x}_i), \tag{23}$$

where $\widehat{u}_\phi(\cdot, \mathbf{z})$ is a differentiable predictor (e.g., GI-TRANSOLVER) and $\psi$ is a scalar loss. The pipeline maintains feasibility via a (possibly iterative) reprojection operator

$$\mathbf{X}(\mathbf{z}) \in \Gamma(\mathbf{z})^N, \qquad \mathbf{X}_{t+1} = \Pi(\mathbf{X}_t, \mathbf{z}_{t+1}). \tag{24}$$

Define the reduced objective

$$\mathcal{J}(\mathbf{z}) \triangleq \widehat{\mathcal{J}}(\mathbf{z}, \mathbf{X}(\mathbf{z})).$$

**Full gradient vs. detached gradient.**

**Proposition A.6 (Full gradient decomposition).** *Assume $\widehat{\mathcal{J}}$ is differentiable and $\mathbf{X}(\cdot)$ is differentiable at $\mathbf{z}$. Then*

$$\nabla_{\mathbf{z}}\mathcal{J}(\mathbf{z}) = \nabla_{\mathbf{z}}\widehat{\mathcal{J}}(\mathbf{z}, \mathbf{X}(\mathbf{z})) + \left(\nabla_{\mathbf{X}}\widehat{\mathcal{J}}(\mathbf{z}, \mathbf{X}(\mathbf{z}))\right)\frac{d\mathbf{X}(\mathbf{z})}{d\mathbf{z}}, \tag{25}$$

*where $\nabla_{\mathbf{z}}\widehat{\mathcal{J}}$ denotes the partial gradient holding $\mathbf{X}$ fixed, and $\nabla_{\mathbf{X}}\widehat{\mathcal{J}}$ is the gradient w.r.t. the stacked variable $\mathbf{X} \in \mathbb{R}^{3N}$.*

*Proof.* Apply the chain rule to $\mathcal{J}(\mathbf{z}) = \widehat{\mathcal{J}}(\mathbf{z}, \mathbf{X}(\mathbf{z}))$. $\qquad\square$

Many implementations stop-gradient through $\mathbf{X}(\mathbf{z})$ and use the *detached* direction

$$\mathbf{g}_{\mathrm{det}}(\mathbf{z}) \triangleq \nabla_{\mathbf{z}}\widehat{\mathcal{J}}(\mathbf{z}, \mathbf{X}(\mathbf{z})). \tag{26}$$

**Bounding the dropped "transport" term via implicit sensitivity**   We reuse Assumption A.3 from Appendix A.2.

**Lemma A.7 (Per-point surface sensitivity bound).** *Under Assumption A.3, for any $\mathbf{z} \in \mathcal{Z}$ and any $\mathbf{x} \in \Gamma(\mathbf{z})$, there exists a locally feasible selection $\mathbf{x}(\cdot)$ with $\mathbf{x}(\mathbf{z}) = \mathbf{x}$ and $\mathbf{x}(\mathbf{z}') \in \Gamma(\mathbf{z}')$ for $\mathbf{z}'$ near $\mathbf{z}$ such that*

$$\left\|\frac{d\mathbf{x}(\mathbf{z})}{d\mathbf{z}}\right\|_{\mathrm{op}} \leq \frac{\|\nabla_{\mathbf{z}}s_\theta(\mathbf{x}, \mathbf{z})\|_2}{\|\nabla_{\mathbf{x}}s_\theta(\mathbf{x}, \mathbf{z})\|_2} \leq \frac{L_{\mathbf{z}}}{m}.$$

*Proof.* Differentiate the implicit constraint $s_\theta(\mathbf{x}(\mathbf{z}), \mathbf{z}) = 0$:

$$\nabla_{\mathbf{x}}s_\theta(\mathbf{x}(\mathbf{z}), \mathbf{z})\frac{d\mathbf{x}(\mathbf{z})}{d\mathbf{z}} + \nabla_{\mathbf{z}}s_\theta(\mathbf{x}(\mathbf{z}), \mathbf{z}) = \mathbf{0}.$$

Taking the minimum-norm solution gives

$$\frac{d\mathbf{x}}{d\mathbf{z}} = -(\nabla_{\mathbf{x}}s_\theta)^\dagger \nabla_{\mathbf{z}}s_\theta.$$

Since $\|\nabla_{\mathbf{x}}s_\theta\|_2 \geq m$, we have $\|(\nabla_{\mathbf{x}}s_\theta)^\dagger\|_{\mathrm{op}} = 1/\|\nabla_{\mathbf{x}}s_\theta\|_2$, yielding

$$\left\|\frac{d\mathbf{x}}{d\mathbf{z}}\right\|_{\mathrm{op}} \leq \|(\nabla_{\mathbf{x}}s_\theta)^\dagger\|_{\mathrm{op}} \|\nabla_{\mathbf{z}}s_\theta\|_2 \leq \frac{\|\nabla_{\mathbf{z}}s_\theta(\mathbf{x}, \mathbf{z})\|_2}{\|\nabla_{\mathbf{x}}s_\theta(\mathbf{x}, \mathbf{z})\|_2} \leq \frac{L_{\mathbf{z}}}{m}.$$

$\qquad\square$

**Theorem A.8 (Gradient mismatch bound (detach drops a controlled term)).** *Let $\mathbf{X}(\mathbf{z}) = (\mathbf{x}_1(\mathbf{z}), \ldots, \mathbf{x}_N(\mathbf{z}))$ be a locally feasible selection with each $\mathbf{x}_i(\mathbf{z}) \in \Gamma(\mathbf{z})$ satisfying Lemma A.7. Then*

$$\left\|\nabla_{\mathbf{z}}\mathcal{J}(\mathbf{z}) - \mathbf{g}_{\mathrm{det}}(\mathbf{z})\right\|_2 = \left\|\left(\nabla_{\mathbf{X}}\widehat{\mathcal{J}}(\mathbf{z}, \mathbf{X}(\mathbf{z}))\right)\frac{d\mathbf{X}(\mathbf{z})}{d\mathbf{z}}\right\|_2 \leq \left\|\nabla_{\mathbf{X}}\widehat{\mathcal{J}}(\mathbf{z}, \mathbf{X}(\mathbf{z}))\right\|_{\mathrm{op}} \left\|\frac{d\mathbf{X}(\mathbf{z})}{d\mathbf{z}}\right\|_{\mathrm{op}}. \tag{27}$$

*Moreover, if $\|d\mathbf{x}_i/d\mathbf{z}\|_{\mathrm{op}} \leq L_{\mathbf{z}}/m$ for all $i$, then (viewing $\mathbf{X}$ as stacked in $\mathbb{R}^{3N}$)*

$$\left\|\frac{d\mathbf{X}(\mathbf{z})}{d\mathbf{z}}\right\|_{\mathrm{op}} \leq \sqrt{N}\frac{L_{\mathbf{z}}}{m}, \qquad \Rightarrow \qquad \left\|\nabla_{\mathbf{z}}\mathcal{J}(\mathbf{z}) - \mathbf{g}_{\mathrm{det}}(\mathbf{z})\right\|_2 \leq \sqrt{N}\frac{L_{\mathbf{z}}}{m}\left\|\nabla_{\mathbf{X}}\widehat{\mathcal{J}}(\mathbf{z}, \mathbf{X}(\mathbf{z}))\right\|_{\mathrm{op}}. \tag{28}$$

*Proof.* Eq. 27 follows from Eq. 25–Eq. 26 and $\|\mathbf{AB}\|_2 \leq \|\mathbf{A}\|_{\mathrm{op}} \|\mathbf{B}\|_{\mathrm{op}}$.

It remains to prove the bound $\|d\mathbf{X}/d\mathbf{z}\|_{\mathrm{op}} \leq \sqrt{N}(L_{\mathbf{z}}/m)$ under $\|d\mathbf{x}_i/d\mathbf{z}\|_{\mathrm{op}} \leq L_{\mathbf{z}}/m$. View $d\mathbf{X}/d\mathbf{z}$ as the stacked linear map $\mathbf{v} \mapsto \big((d\mathbf{x}_1/d\mathbf{z})\mathbf{v}, \ldots, (d\mathbf{x}_N/d\mathbf{z})\mathbf{v}\big) \in \mathbb{R}^{3N}$. Then for any $\mathbf{v} \in \mathbb{R}^d$,

$$\left\| \frac{d\mathbf{X}(\mathbf{z})}{d\mathbf{z}}\mathbf{v} \right\|_2^2 = \sum_{i=1}^N \left\| \frac{d\mathbf{x}_i(\mathbf{z})}{d\mathbf{z}}\mathbf{v} \right\|_2^2 \leq \sum_{i=1}^N \left\| \frac{d\mathbf{x}_i(\mathbf{z})}{d\mathbf{z}} \right\|_{\mathrm{op}}^2 \|\mathbf{v}\|_2^2 \leq N \left( \frac{L_{\mathbf{z}}}{m} \right)^2 \|\mathbf{v}\|_2^2.$$

Taking supremum over $\mathbf{v} \neq \mathbf{0}$ yields

$$\left\| \frac{d\mathbf{X}(\mathbf{z})}{d\mathbf{z}} \right\|_{\mathrm{op}} = \sup_{\mathbf{v}\neq\mathbf{0}} \frac{\|(d\mathbf{X}/d\mathbf{z})\mathbf{v}\|_2}{\|\mathbf{v}\|_2} \leq \sqrt{N}\,\frac{L_{\mathbf{z}}}{m}.$$

Substituting into Eq. 27 gives Eq. 28. $\qquad\square$

**Corollary A.9** (Why STABLESDF helps detached optimization). *If training reduces $\sup_{\mathbf{x}\in N_\rho(\Gamma(\mathbf{z}))} \|\nabla_{\mathbf{z}} s_\theta(\mathbf{x}, \mathbf{z})\|_2$ on a neighborhood of the surface (thus reducing $L_{\mathbf{z}}$ in Assumption A.3(2)), then the mismatch bound Eq. 28 tightens proportionally, making detached gradients closer to the full reduced gradient and improving stability of latent shape optimization.*

### A.4. Null-space projection in latent space

**Overview.** This section establishes a projection mechanism to preserve linearized implicit constraints in the latent space. In particular, the closest feasible update under the linearized constraint $\mathbf{G}(\mathbf{z})\Delta\mathbf{z} = \mathbf{0}$ is given by orthogonal projection onto $\mathrm{Null}(\mathbf{G})$, and we further show first-order constraint invariance under the projected update.

**Setup.** Let $\mathbf{X}_{\mathrm{const}} = \{\mathbf{x}_1, \ldots, \mathbf{x}_M\} \subset \mathbb{R}^3$ be protected query points. Define the constraint vector and its Jacobian

$$\mathbf{c}(\mathbf{z}) \triangleq \begin{bmatrix} s_\theta(\mathbf{x}_1, \mathbf{z}) \\ \vdots \\ s_\theta(\mathbf{x}_M, \mathbf{z}) \end{bmatrix} \in \mathbb{R}^M, \qquad \mathbf{G}(\mathbf{z}) \triangleq \nabla_{\mathbf{z}}\mathbf{c}(\mathbf{z}) \in \mathbb{R}^{M\times d}.$$

Given a proposed direction $\mathbf{g} \in \mathbb{R}^d$, we enforce first-order constraint preservation: $\mathbf{c}(\mathbf{z} + \Delta\mathbf{z}) \approx \mathbf{c}(\mathbf{z}) \iff \mathbf{G}(\mathbf{z})\Delta\mathbf{z} = \mathbf{0}$.

**Main theorem.**

**Theorem A.10** (Optimality of null-space projection (closest feasible update)). *Fix $\mathbf{z}$ and write $\mathbf{G} \triangleq \mathbf{G}(\mathbf{z})$. Consider*

$$\Delta\mathbf{z}^\star \in \arg\min_{\Delta\mathbf{z}\in\mathbb{R}^d} \|\Delta\mathbf{z} - \mathbf{g}\|_2^2 \quad s.t. \quad \mathbf{G}\Delta\mathbf{z} = \mathbf{0}.$$

*Then one optimal solution is*

$$\Delta\mathbf{z}^\star = \mathbf{P}(\mathbf{z})\,\mathbf{g}, \qquad \mathbf{P}(\mathbf{z}) \triangleq \mathbf{I}_d - \mathbf{G}(\mathbf{z})^\dagger \mathbf{G}(\mathbf{z}),$$

*where $(\cdot)^\dagger$ is the Moore–Penrose pseudoinverse. Moreover, $\mathbf{P}(\mathbf{z})$ is the orthogonal projector onto $\mathrm{Null}(\mathbf{G}(\mathbf{z}))$, i.e.,*

$$\mathbf{P}(\mathbf{z})^2 = \mathbf{P}(\mathbf{z}), \qquad \mathbf{P}(\mathbf{z})^\top = \mathbf{P}(\mathbf{z}), \qquad \mathbf{G}(\mathbf{z})\mathbf{P}(\mathbf{z}) = \mathbf{0}.$$

*Proof.* Introduce the Lagrangian
$$\mathcal{L}(\Delta\mathbf{z}, \boldsymbol{\lambda}) = \|\Delta\mathbf{z} - \mathbf{g}\|_2^2 + 2\boldsymbol{\lambda}^\top(\mathbf{G}\Delta\mathbf{z}).$$

Stationarity gives $2(\Delta\mathbf{z} - \mathbf{g}) + 2\mathbf{G}^\top\boldsymbol{\lambda} = \mathbf{0}$, hence $\Delta\mathbf{z} = \mathbf{g} - \mathbf{G}^\top\boldsymbol{\lambda}$. Imposing $\mathbf{G}\Delta\mathbf{z} = \mathbf{0}$ yields $(\mathbf{GG}^\top)\boldsymbol{\lambda} = \mathbf{Gg}$. In the general (possibly rank-deficient) case, the minimum-norm solution is $\boldsymbol{\lambda} = (\mathbf{GG}^\top)^\dagger\mathbf{Gg}$, giving

$$\Delta\mathbf{z}^\star = \mathbf{g} - \mathbf{G}^\top(\mathbf{GG}^\top)^\dagger\mathbf{Gg}.$$

Using the identity $\mathbf{G}^\dagger = \mathbf{G}^\top(\mathbf{GG}^\top)^\dagger$, we obtain $\Delta\mathbf{z}^\star = (\mathbf{I}_d - \mathbf{G}^\dagger\mathbf{G})\mathbf{g} = \mathbf{P}(\mathbf{z})\mathbf{g}$. The projector properties follow from standard Moore–Penrose relations: $\mathbf{GG}^\dagger\mathbf{G} = \mathbf{G}$ implies $\mathbf{GP} = \mathbf{0}$; symmetry $(\mathbf{G}^\dagger\mathbf{G})^\top = \mathbf{G}^\dagger\mathbf{G}$ implies $\mathbf{P}^\top = \mathbf{P}$; and $(\mathbf{G}^\dagger\mathbf{G})^2 = \mathbf{G}^\dagger\mathbf{G}$ implies $\mathbf{P}^2 = \mathbf{P}$. $\qquad\square$

**Corollary A.11** (First-order invariance under projected updates). *Let $\mathbf{z}^+ = \mathbf{z} - \eta\mathbf{P}(\mathbf{z})\mathbf{g}$ for any $\mathbf{g} \in \mathbb{R}^d$ and $\eta > 0$. Then, as $\eta \to 0$,*

$$\mathbf{c}(\mathbf{z}^+) = \mathbf{c}(\mathbf{z}) + \mathcal{O}(\eta^2).$$

*Proof.* By Taylor expansion, $\mathbf{c}(\mathbf{z}^+) = \mathbf{c}(\mathbf{z}) + \mathbf{G}(\mathbf{z})(\mathbf{z}^+ - \mathbf{z}) + \mathcal{O}(\|\mathbf{z}^+ - \mathbf{z}\|_2^2) = \mathbf{c}(\mathbf{z}) - \eta\mathbf{G}(\mathbf{z})\mathbf{P}(\mathbf{z})\mathbf{g} + \mathcal{O}(\eta^2)$. Since $\mathbf{G}(\mathbf{z})\mathbf{P}(\mathbf{z}) = \mathbf{0}$, the first-order term vanishes. $\qquad\qquad\square$

### A.5. Remeshing-free projection in spatial domain

**Overview.** This section justifies the remeshing-free reprojection step that maps off-surface points back to the implicit zero level set. We show it can be interpreted as a Gauss–Newton step for minimizing the squared level-set residual, and establish local quadratic residual contraction and a distance-to-surface bound under a regular level-set assumption.

After each latent update, the implicit surface $\Gamma(\mathbf{z}) \triangleq \{\mathbf{x} \in \mathbb{R}^3 : s_\theta(\mathbf{x}, \mathbf{z}) = 0\}$ changes, and previously sampled boundary queries may drift off the updated boundary. We therefore reproject each query point back to the zero level set using the update

$$\mathbf{x}^+ \leftarrow \mathbf{x} - s_\theta(\mathbf{x}, \mathbf{z}) \frac{\nabla_{\mathbf{x}} s_\theta(\mathbf{x}, \mathbf{z})}{\|\nabla_{\mathbf{x}} s_\theta(\mathbf{x}, \mathbf{z})\|_2^2 + \varepsilon}, \tag{29}$$

where $\varepsilon > 0$ is a small constant for numerical stability. In this appendix, we justify Eq. 29 as (i) a Gauss–Newton step for a natural projection objective, and (ii) a locally convergent root-finding method that drives points to the boundary efficiently.

**Setup.** Fix a latent code $\mathbf{z}$ and define the scalar field

$$g(\mathbf{x}) \triangleq s_\theta(\mathbf{x}, \mathbf{z}), \qquad \Gamma \triangleq \{\mathbf{x} : g(\mathbf{x}) = 0\}.$$

We assume $g$ is twice continuously differentiable in a tubular neighborhood of $\Gamma$.

**Assumption A.12** (Regular level set and bounded curvature). *There exist constants $m > 0$, $M < \infty$, and $\rho > 0$ such that for all $\mathbf{x} \in N_\rho(\Gamma) \triangleq \{\mathbf{x} : \mathrm{dist}(\mathbf{x}, \Gamma) \leq \rho\}$:*

$$\|\nabla_{\mathbf{x}} g(\mathbf{x})\|_2 \geq m, \qquad \|\nabla_{\mathbf{xx}}^2 g(\mathbf{x})\|_{\mathrm{op}} \leq M. \tag{30}$$

**Eq. 29 as a Gauss–Newton step** A natural way to reproject a point $\mathbf{x}$ onto $\Gamma$ is to minimize the squared level-set residual:

$$\min_{\mathbf{y} \in \mathbb{R}^3} \phi(\mathbf{y}) \quad \text{where} \quad \phi(\mathbf{y}) \triangleq \frac{1}{2} g(\mathbf{y})^2. \tag{31}$$

We have $\nabla_{\mathbf{x}} \phi(\mathbf{x}) = g(\mathbf{x}) \nabla_{\mathbf{x}} g(\mathbf{x})$ and

$$\nabla_{\mathbf{xx}}^2 \phi(\mathbf{x}) = \nabla_{\mathbf{x}} g(\mathbf{x}) \nabla_{\mathbf{x}} g(\mathbf{x})^\top + g(\mathbf{x}) \nabla_{\mathbf{xx}}^2 g(\mathbf{x}).$$

The Gauss–Newton method ignores the second term and uses the approximate Hessian $\mathbf{B}(\mathbf{x}) = \nabla_{\mathbf{x}} g(\mathbf{x}) \nabla_{\mathbf{x}} g(\mathbf{x})^\top$. The Gauss–Newton step $\Delta$ solves

$$\mathbf{B}(\mathbf{x})\Delta = -\nabla_{\mathbf{x}} \phi(\mathbf{x}) \quad \Longleftrightarrow \quad (\nabla_{\mathbf{x}} g \nabla_{\mathbf{x}} g^\top)\Delta = -g \nabla_{\mathbf{x}} g.$$

Since $\mathbf{B}(\mathbf{x})$ is rank-1, the minimum-norm solution lies in the span of $\nabla_{\mathbf{x}} g(\mathbf{x})$ and is given by

$$\Delta_{\mathrm{GN}} = -g(\mathbf{x}) \frac{\nabla_{\mathbf{x}} g(\mathbf{x})}{\|\nabla_{\mathbf{x}} g(\mathbf{x})\|_2^2}. \tag{32}$$

Thus, when $\varepsilon = 0$, Eq. 29 is exactly one Gauss–Newton step for minimizing $\frac{1}{2} g(\mathbf{x})^2$, i.e., it moves along the local normal direction and cancels the first-order residual.

**Numerical stabilization.** With $\varepsilon > 0$, Eq. 29 is a damped variant of Eq. 32:

$$\Delta_\varepsilon = -g(\mathbf{x}) \frac{\nabla_{\mathbf{x}} g(\mathbf{x})}{\|\nabla_{\mathbf{x}} g(\mathbf{x})\|_2^2 + \varepsilon},$$

which prevents overly large steps when $\|\nabla_{\mathbf{x}} g\|_2$ is small (rare under Assumption A.12 but helpful in practice).

**Local residual reduction and convergence.** We now show that the reprojection step exhibits *quadratic residual reduction* under Assumption A.12.

**Proposition A.13** (One-step residual contraction (quadratic)). *Under Assumption A.12, fix any* $\mathbf{x} \in N_\rho(\Gamma)$ *and define the update*

$$\mathbf{x}^+ \triangleq \mathbf{x} - g(\mathbf{x}) \frac{\nabla_\mathbf{x} g(\mathbf{x})}{\|\nabla_\mathbf{x} g(\mathbf{x})\|_2^2}. \tag{33}$$

*Then the step length and the post-update residual satisfy*

$$\|\mathbf{x}^+ - \mathbf{x}\|_2 \leq \frac{|g(\mathbf{x})|}{m}, \qquad |g(\mathbf{x}^+)| \leq \frac{M}{2m^2} |g(\mathbf{x})|^2. \tag{34}$$

*Consequently, if* $|g(\mathbf{x})|$ *is sufficiently small, repeated application of Eq. 33 drives* $\mathbf{x}$ *to* $\Gamma$ *rapidly (quadratic decrease of* $|g|$*).*

*Proof.* Let $\Delta \triangleq \mathbf{x}^+ - \mathbf{x} = -g(\mathbf{x}) \nabla_\mathbf{x} g(\mathbf{x})/\|\nabla_\mathbf{x} g(\mathbf{x})\|_2^2$. The step length bound follows immediately:

$$\|\Delta\|_2 = |g(\mathbf{x})| \frac{\|\nabla_\mathbf{x} g(\mathbf{x})\|_2}{\|\nabla_\mathbf{x} g(\mathbf{x})\|_2^2} = \frac{|g(\mathbf{x})|}{\|\nabla_\mathbf{x} g(\mathbf{x})\|_2} \leq \frac{|g(\mathbf{x})|}{m}.$$

For the residual, apply the second-order Taylor expansion of $g$ at $\mathbf{x}$: there exists $\tau \in (0, 1)$ such that

$$g(\mathbf{x} + \Delta) = g(\mathbf{x}) + \nabla_\mathbf{x} g(\mathbf{x})^\top \Delta + \frac{1}{2} \Delta^\top \nabla_{\mathbf{xx}}^2 g(\mathbf{x} + \tau\Delta)\Delta.$$

By construction,

$$\nabla_\mathbf{x} g(\mathbf{x})^\top \Delta = -g(\mathbf{x}) \frac{\|\nabla_\mathbf{x} g(\mathbf{x})\|_2^2}{\|\nabla_\mathbf{x} g(\mathbf{x})\|_2^2} = -g(\mathbf{x}),$$

so the zeroth and first-order terms cancel, yielding

$$g(\mathbf{x}^+) = \frac{1}{2} \Delta^\top \nabla_{\mathbf{xx}}^2 g(\mathbf{x} + \tau\Delta)\Delta.$$

Taking absolute values and using $\|\nabla_{\mathbf{xx}}^2 g\|_{\mathrm{op}} \leq M$ (Assumption A.12),

$$|g(\mathbf{x}^+)| \leq \frac{1}{2} \|\nabla_{\mathbf{xx}}^2 g(\mathbf{x} + \tau\Delta)\|_{\mathrm{op}} \|\Delta\|_2^2 \leq \frac{M}{2} \|\Delta\|_2^2 \leq \frac{M}{2} \left(\frac{|g(\mathbf{x})|}{m}\right)^2 = \frac{M}{2m^2} |g(\mathbf{x})|^2,$$

which proves Eq. 34. $\qquad\square$

**Distance-to-surface bound (why the step is a projection).** The previous result shows that the update reduces the level-set residual quickly. We also provide a geometric bound connecting $|g(\mathbf{x})|$ to the Euclidean distance to the surface.

**Lemma A.14** (Residual controls distance to $\Gamma$). *Under Assumption A.12, for any* $\mathbf{x} \in N_\rho(\Gamma)$*,*

$$\mathrm{dist}(\mathbf{x}, \Gamma) \leq \frac{|g(\mathbf{x})|}{m}. \tag{35}$$

*Proof.* Consider the curve $\mathbf{x}(t) = \mathbf{x} - t \nabla_\mathbf{x} g(\mathbf{x})/\|\nabla_\mathbf{x} g(\mathbf{x})\|_2$ for $t \geq 0$. Define $h(t) \triangleq g(\mathbf{x}(t))$. Then $h(0) = g(\mathbf{x})$ and

$$h'(0) = \nabla_\mathbf{x} g(\mathbf{x})^\top \left(-\frac{\nabla_\mathbf{x} g(\mathbf{x})}{\|\nabla_\mathbf{x} g(\mathbf{x})\|_2}\right) = -\|\nabla_\mathbf{x} g(\mathbf{x})\|_2 \leq -m.$$

By continuity, for $\mathbf{x} \in N_\rho(\Gamma)$ there exists a small $t^\star \geq 0$ such that $h(t^\star) = 0$, i.e., $\mathbf{x}(t^\star) \in \Gamma$, and moreover $t^\star \leq |g(\mathbf{x})|/m$ by the mean value theorem applied to $h$ on $[0, t^\star]$. Therefore $\mathrm{dist}(\mathbf{x}, \Gamma) \leq \|\mathbf{x} - \mathbf{x}(t^\star)\|_2 = t^\star \leq |g(\mathbf{x})|/m$, proving Eq. 35. $\qquad\square$

*Table 4.* Comparison with prior works on geometry modeling, physics prediction, and optimization (Part I). We unify terminology: *Geometry rep.* refers to the representation used to reconstruct the shape (implicit SDF or voxel occupancy); *Latent rep.* denotes the parameterization of the design space.

| Method | Geometry representation | Physics prediction | Optimization | Remeshing-free | Flexible objective | Controllable optimization |
|---|---|---|---|---|---|---|
| PHYSGEN (You et al., 2025) | **VAE** with unified **physics–geometry** latents; encode **point cloud** to **high-dim vector**; decoder predicts **implicit SDF** | Physics decoder predicts **surface pressure** ($p$) and **global drag** ($C_D$) | Objective-guided **flow-matching** sampling with $C_D$-gradient guidance; then local "**physics refinement**" | No | No | No |
| 3DID (Hao et al., 2025) | **VAE** with unified **physics-geometry** latents; encode **point cloud + *physical field*** to **tri-plane**; decoder predicts **voxel occupancy** | Stage 1: MLP predicts **global drag** ($C_D$) and **pressure** on voxel vertex; Stage 2: MGN predicts pressure on control points | Objective-guided **diffusion** sampling with $C_D$ gradient; then **refinement** via **FFD** | No | No | No |
| AGML (Tran et al., 2024) | **AE** with **physics-geometry** latent space; encode **voxel** to **3D tensor**; decoder predicts **voxel occupancy** | Drag decoder predicts **global drag** ($C_D$) | Latent-space **gradient descent** by backpropagation through drag decoder | Yes | No | No |
| TRIPOPTIMIZER (Vatani et al., 2025) | **VAE** with unified **physics-geometry** latents; encode **point cloud** to **tri-plane**; decoder predicts **semi-continuous occupancy** | Drag decoder predicts **global drag** ($C_D$) | "**Encoder fine-tuning**" optimization: keep input point cloud fixed, fine-tune part of the encoder to change latent code | Yes | No | No |
| GANO (ours) | **Auto-decoder** learns an **independent** geometry latent; encode **point cloud** to **256-dim vector**; decoder predicts **implicit SDF** | GI-TRANSOLVER predicts **surface pressure** ($p$) | Latent-space **gradient descent**: backpropagation through a pressure-integral drag surrogate | Yes | **Yes** | **Yes** |

# B. Detailed Comparison with Related Works

In this section, we review representative latent-space optimization models. We provide a detailed summary of these methods from three aspects, including geometric representation, physics prediction, and optimization strategy (Table 4).

**Geometric representation.** **3DID (Hao et al., 2025)** and **AGML (Tran et al., 2024)** adopt voxel occupancy, while **TripOptimizer (Vatani et al., 2025)** uses a semi-continuous occupancy representation. Voxel-based representations typically suffer from *limited fidelity*: if the initial shape to be optimized cannot be faithfully represented, subsequent optimization *lacks a reliable foundation*. Optimizing a geometry that already deviates from the true initial shape may yield a numerically favorable solution that nevertheless deviates from the true design intent, and thus *may be of limited practical value*. Moreover, converting voxels into smooth surfaces still requires *additional* manual post-processing. In contrast, both **PhysGen (You et al., 2025)** and our **GANO** use an *implicit SDF field* to represent geometry accurately and efficiently, achieving *excellent* reconstruction quality (Table 7; *our F1 score exceeds 0.90*). It is also worth noting that **3DID** *additionally requires physical-field inputs of the initial geometry*, and therefore cannot avoid relying on conventional CFD simulations. Above mentioned methods (Tran et al., 2024; Vatani et al., 2025; Hao et al., 2025; You et al., 2025) entangle geometric and physical information into a single latent code and jointly train a geometry decoder head and a physics decoder head. In practice, balancing these two losses involves *highly sensitive hyperparameters*. We argue that such entanglement can lead to information interference, thereby degrading both geometric reconstruction and physics prediction accuracy. Furthermore, overly *high-dimensional latent* representations like tri-plane tend to make optimization *unstable*. To address these issues, we *decouple geometric representation from physics prediction*, enabling both strong geometry reconstruction and accurate physics prediction. Meanwhile, we can still *manipulate geometry* without sacrificing physical information via a *null-space projection*. Finally, a relatively *low* latent dimensionality (256) makes the optimization domain *continuous and compact*, further improving stability.

**Physics prediction.** **TripOptimizer** and **AGML** optimize geometry using only *scalar objectives*. While **3DID** and **PhysGen** incorporate *physical-field information*, they rely on existing surrogates and still need *reconstruction or remeshing* after each update, which introduces significant computational overhead. In contrast, we achieve state-of-the-art physics prediction via geometric injection, which helps produce high-quality optimization outcomes.

**Optimization strategy.** **3DID** and **PhysGen** employ *diffusion/flow-based* generative models. However, they suffer from the

inaccuracy predicting drag from a noisy latent code, which can cause bias in posterior sampling. Moreover, both methods require a *two-stage post-processing pipeline*, which undermines the simplicity of the overall optimization procedure. **AGML** and **TripOptimizer**, on the other hand, *directly update design variables via gradient descent*; although faster, this often leads to *unstable* optimization like OOD deformation. Our method instead leverages StableSDF, an auto-decoder geometry representation model that *retains generative capability without diffusion sampling* (Fig. 3**b**). Thus we predict pressure based on noise-free and meaningful latent code avoiding inaccurate estimation of $\mathbb{E}[\mathbf{z_0}|\mathbf{z_t}]$. The incorporated denoising mechanism further makes the optimization process stable and continuous. In addition, our remeshing-free projection allows us to predict the physical field after geometry updates *without* explicitly reconstructing the geometry, substantially improving optimization efficiency.

## C. Algorithm Details

We provide a detailed description of the optimization algorithm here.

Alg. 1 presents the procedure of using GANO to optimize a car shape. We select a car from DrivAerNet++ and encode it into a latent code $\mathbf{z}_0$. Car optimization requires the surface point cloud, which will be updated during optimization.

Alg. 2 presents the procedure of using GANO to optimize an airfoil shape. We select an airfoil from AirFoil_9k and encode it into a latent code $\mathbf{z}_0$. Airfoil optimization requires the mesh around the airfoil, which will be updated during optimization.

---

**Algorithm 1** GANO Optimize vehicle

---

**Input:** initial latent code $\mathbf{z}_0$; initial surface samples $\mathcal{X}_0 = \{\mathbf{x}_i\}_{i=1}^N \subset \Gamma(\mathbf{z}_0)$; geometry decoder $s_\theta$ (STABLESDF); field predictor $f_\phi$ (GI-TRANSOLVER); (optional) constraint points $\mathcal{X}_{\text{const}}$
**Output:** optimized latent code $\mathbf{z}^\star$; boundary-consistent samples $\mathcal{X}^\star$
$\mathbf{z} \leftarrow \mathbf{z}_0, \ \mathcal{X} \leftarrow \mathcal{X}_0$
**for** $t = 1$ **to** $T$ **do**
  **(A) Predict pressure on vehicle surface**
  Construct query set $\mathcal{Q} \leftarrow \mathcal{Q}(\mathcal{X})$
  Predict pressure $\hat{\mathbf{p}} \leftarrow f_\phi(\mathcal{Q}, \mathbf{z})$
  **(B) Objective + latent update**
  Compute objective $\mathcal{J}(\mathbf{z})$
  Backpropagate $\mathbf{g} \leftarrow \nabla_{\mathbf{z}} \mathcal{J}(\mathbf{z})$
  **if** $\mathcal{X}_{\text{const}}$ is provided **then**
    $\mathbf{G} \leftarrow \frac{\partial s_\theta(\mathcal{X}_{\text{const}}, \mathbf{z})}{\partial \mathbf{z}}$
    $\mathbf{g} \leftarrow (\mathbf{I} - \mathbf{G}^\dagger \mathbf{G})\mathbf{g}$       (null-space projection)
  **end if**
  Update latent code $\mathbf{z} \leftarrow \text{Adam}(\mathbf{z}, \mathbf{g})$
  **(C) Remeshing-free projection**
  **for** $k = 1$ **to** $K$ **do**
    **for** $i = 1$ **to** $N$ **do**
      $\mathbf{x}_i \leftarrow \mathbf{x}_i - s_\theta(\mathbf{x}_i, \mathbf{z}) \frac{\nabla_{\mathbf{x}} s_\theta(\mathbf{x}_i, \mathbf{z})}{\|\nabla_{\mathbf{x}} s_\theta(\mathbf{x}_i, \mathbf{z})\|_2^2 + \varepsilon}$
    **end for**
  **end for**
**end for**
$\mathbf{z}^\star \leftarrow \mathbf{z}, \ \mathcal{X}^\star \leftarrow \mathcal{X}$

---

---

**Algorithm 2** GANO Optimize airfoil

---

**Input:** initial latent $\mathbf{z}_0$; initial meshes $\mathcal{X}_0 = \{\mathbf{x}_i\}_{i=1}^{N} \subset \mathbb{R}^2$ ; geometry decoder $s_\theta$ (STABLESDF); physics predictor $f_\phi$ (GI-TRANSOLVER);

      control-volume boundary points $\mathcal{B} = \{\mathbf{b}_j\}_{j=1}^{N_b}$ with slicing $\{\mathcal{B}_L, \mathcal{B}_R, \mathcal{B}_B, \mathcal{B}_T\}$ and spacings $(\Delta x, \Delta y)$;

      freestream $(\rho, p_\infty, q_\infty, \alpha)$; constraint $C_d^{\max}$; weights $\lambda_{cd}, \lambda_{reg}$.

**Output:** optimized latent $\mathbf{z}^\star$; boundary-consistent meshes $\mathcal{X}^\star$;

Reference SDF values $\mathbf{d}_{\mathrm{ref}} \leftarrow s_\theta(\mathcal{X}_0, \mathbf{z}_0)$

$\mathbf{z} \leftarrow \mathbf{z}_0, \quad \mathcal{X} \leftarrow \mathcal{X}_0$

**for** $t = 1$ **to** $T$ **do**

   **(A) Predict flow on CV boundary + compute aerodynamics**

   Construct query set $\mathcal{Q} \leftarrow \mathcal{X} \cup \mathcal{B}$

   Predict fields $\mathbf{y} \leftarrow f_\phi(\mathcal{Q}, \mathbf{z}), \quad$ extract $(u, v, p)$

   Gauge pressure $p_g \leftarrow p - p_\infty$

   Compute forces $(L, D) \leftarrow \mathrm{CVForces}(\mathcal{B}_L, \mathcal{B}_R, \mathcal{B}_B, \mathcal{B}_T; u, v, p_g, \Delta x, \Delta y, \rho, \alpha)$

   $C_\ell \leftarrow \frac{L}{q_\infty c}, \quad C_d \leftarrow \frac{D}{q_\infty c}$

   **(B) Objective + latent update**

   Penalty $r \leftarrow \mathrm{ReLU}(C_d - C_d^{\max})$

   $\mathcal{J}(\mathbf{z}) \leftarrow -C_\ell + \lambda_{cd}\, r^2$

   Backpropagate $\mathbf{g} \leftarrow \nabla_{\mathbf{z}} \mathcal{J}(\mathbf{z})$

   $\mathbf{z} \leftarrow \mathrm{Adam}(\mathbf{z}, \mathbf{g})$

   **(C) Remeshing-free projection**

   **for** $k = 1$ **to** $K$ **do**

     **for** $i = 1$ **to** $N$ **do**

       $d \leftarrow s_\theta(\mathbf{x}_i, \mathbf{z}), \quad \mathbf{g}_x \leftarrow \nabla_{\mathbf{x}} s_\theta(\mathbf{x}_i, \mathbf{z})$

       $\mathbf{x}_i \leftarrow \mathbf{x}_i - (d - \mathbf{d}_{\mathrm{ref},i}) \frac{\mathbf{g}_x}{\|\mathbf{g}_x\|_2^2 + \varepsilon}$

     **end for**

   **end for**

**end for**

$\mathbf{z}^\star \leftarrow \mathbf{z}, \quad \mathcal{X}^\star \leftarrow \mathcal{X}$

---

# D. Experimental Details

Here we describe the experimental details, including the benchmarks, baselines, evaluation metrics, and more.

## D.1. Benchmarks

*Table 5.* Dataset summary.

| Dataset | Dim. | Regular Mesh | Dataset Size | #Points |
|---------|------|--------------|--------------|---------|
| Helmholtz | 2D | YES | 1000 shapes $\times$ 10 angles | 65536 |
| AirFoil_9k | 2D | NO | 8996 | $\approx 50000$ |
| DrivAerNet++ | 3D | NO | 8000 | $\approx 500000$ |

**Helmholtz Inverse Scattering: Experimental Setup.** To evaluate the model capability on a 2D uniform grid, we consider a two-dimensional Helmholtz inverse scattering problem with a penetrable obstacle. In $\mathbb{R}^2$, the total field $\psi^{\text{tot}}$ satisfies

$$\Delta\psi^{\text{tot}}(\mathbf{x}) + \kappa^2\,\varepsilon(\mathbf{x})\,\psi^{\text{tot}}(\mathbf{x}) = 0, \qquad \mathbf{x} \in \mathbb{R}^2, \tag{36}$$

where $\kappa > 0$ is the wavenumber and $\varepsilon(\mathbf{x})$ denotes the relative permittivity. We set

$$\varepsilon(\mathbf{x}) = 1 + q(\mathbf{x}), \qquad q(\mathbf{x}) > -1, \tag{37}$$

and assume that $q$ has compact support, whose support region corresponds to the obstacle shape to be reconstructed. The incident plane wave is defined as

$$\psi^{\text{inc}}(\mathbf{x}; \kappa, \mathbf{d}) = e^{i\kappa\mathbf{x}\cdot\mathbf{d}}, \qquad \mathbf{d} \in \mathbb{S}^1, \tag{38}$$

and we decompose the total field as

$$\psi^{\text{tot}} = \psi^{\text{inc}} + \psi, \tag{39}$$

where $\psi$ denotes the scattered field. The scattered field satisfies

$$\Delta\psi(\mathbf{x}) + \kappa^2(1 + q(\mathbf{x}))\,\psi(\mathbf{x}) = -\kappa^2 q(\mathbf{x})\,\psi^{\text{inc}}(\mathbf{x}), \qquad \mathbf{x} \in \mathbb{R}^2, \tag{40}$$

together with the Sommerfeld radiation condition

$$\lim_{\rho\to\infty} \sqrt{\rho}\left(\frac{\partial\psi}{\partial\rho} - i\kappa\psi\right) = 0, \qquad \rho = \|\mathbf{x}\|_2, \tag{41}$$

uniformly in all directions. In practice, the unbounded exterior domain is truncated to a bounded computational domain, and we impose a first-order absorbing boundary condition (first-order ABC) on $\partial\mathcal{D}$ to approximate the radiation condition:

$$\partial_n\psi(\mathbf{x}) - i\kappa\psi(\mathbf{x}) = 0, \qquad \mathbf{x} \in \partial\mathcal{D}. \tag{42}$$

In the inverse problem, we represent the obstacle by the support of $q$, denoted as $\Omega$, and define

$$q(\mathbf{x}) = \begin{cases} \tilde{q}, & \mathbf{x} \in \Omega, \\ 0, & \mathbf{x} \in \mathbb{R}^2 \setminus \Omega, \end{cases} \qquad \tilde{q} > 0, \tag{43}$$

so that the inverse problem is equivalent to recovering the shape of $\Omega$ from boundary observations when $\tilde{q}$ is known. The computational domain is $\mathcal{D} = [-1, 1]^2$. We use a $256 \times 256$ finite-difference method (FDM) solver for the forward problem to generate the dataset. The dataset contains 1000 randomly generated obstacle shapes, each constrained within a circle of radius 0.5 centered at the origin. The obstacle boundary is generated using random Fourier descriptors: in polar coordinates, we parameterize the boundary radius function by random Fourier coefficients of orders 3 to 8, yielding a family of shapes with varying geometric complexity and frequency content. For each shape, we simulate the scattered field under 10 *fixed* incident angles. For the inverse problem, we randomly sample $N_{\text{obs}} = 100$ observation points on an observation circle $\Gamma_{\text{obs}} = \{\mathbf{x} \in \mathbb{R}^2 : \|\mathbf{x}\|_2 = 0.5\}$, and use the complex-valued scattered field at these points as observations

to reconstruct the obstacle geometry. We split the dataset into training/validation/test sets with a ratio of $8 : 1 : 1$. To consistently encode information from different incident angles, we concatenate the incident angle (or equivalently, the incident direction parameter) with the uniform-grid coordinates along the feature dimension, and use it as a conditional input for the forward prediction task. For fair comparisons in the forward task, we additionally concatenate the shape-specific SDF field as a geometric representation input to the baseline models, and report their results under the same training protocol. While classical recursive linearization methods typically require measurements at multiple wavenumbers (multi-$k$) to progressively recover geometric details across frequencies, our approach exhibits shape reconstruction capability under a *single-wavenumber* setting: in our experiments, we only use observations at $k = 7$ to recover the obstacle shape.

**2D Airfoil Flow Prediction and Optimization: AirFoil_9k.** To assess model performance on 2D non-uniform meshes, we conduct a two-dimensional airfoil aerodynamics experiment using the **AirFoil_9k** dataset. This dataset contains 8996 distinct airfoil geometries along with corresponding flow-field data. The airfoil shapes span a wide range of geometries and provide substantial diversity to evaluate both flow prediction and subsequent differentiable, surrogate-based shape optimization. All airfoils are standardized with chord length $c = 1$. Each airfoil geometry is parameterized by a 19-dimensional CST (Class-Shape Transformation) vector. We take the coordinates of mesh points around the airfoil as input and predict the flow-field quantities

$$\mathbf{u}(\mathbf{x}) = (u(\mathbf{x}), v(\mathbf{x}), p(\mathbf{x})),$$

where $\mathbf{x}$ denotes the mesh coordinate, $u$ and $v$ are the horizontal and vertical velocity components, and $p$ is the pressure. We split the dataset into training/validation/test sets with a ratio of $8 : 1 : 1$. All models are trained under the same data split and evaluated on the test set for flow prediction performance. The optimization objective is to maximize the lift coefficient $C_L$ under a given angle of attack and freestream condition, while constraining the drag coefficient $C_D \leq C_D^{\max}$ via a soft constraint. To compute $C_L$ and $C_D$, we use a control-volume (CV) method, discretizing the pressure and momentum fluxes on the CV boundary to obtain the net force components $(F_x, F_y)$. Let $\Omega_{\mathrm{cv}}$ be the CV and $\Gamma = \partial\Omega_{\mathrm{cv}}$ its boundary with outward unit normal $\mathbf{n}$. Denote the velocity field by $\mathbf{u} = (u, v)^\top$ and the *gauge pressure* by $p' = p - p_\infty$. Ignoring viscous traction on a sufficiently far boundary, the net force per unit span acting on the airfoil is

$$\mathbf{F} = - \oint_\Gamma \left( \rho\, \mathbf{u}\, (\mathbf{u} \cdot \mathbf{n}) + p'\, \mathbf{n} \right) ds, \tag{44}$$

where $\mathbf{F} = (F_x, F_y)^\top$. In our implementation, the left/right boundaries are integrated along $y$ with step size $\Delta y$, the top/bottom boundaries are integrated along $x$ with step size $\Delta x$, and we set a constant density $\rho = 1.0$. The corresponding discrete summation is

$$F_x = \sum_y \left( p_g + \rho u^2 \right)\Big|_{x=x_{\min}} \Delta y + \sum_y \left( - p_g - \rho u^2 \right)\Big|_{x=x_{\max}} \Delta y + \sum_x \left( \rho uv \right)\Big|_{y=y_{\min}} \Delta x + \sum_x \left( - \rho uv \right)\Big|_{y=y_{\max}} \Delta x, \tag{45}$$

$$F_y = \sum_y \left( \rho uv \right)\Big|_{x=x_{\min}} \Delta y + \sum_y \left( - \rho uv \right)\Big|_{x=x_{\max}} \Delta y + \sum_x \left( p_g + \rho v^2 \right)\Big|_{y=y_{\min}} \Delta x + \sum_x \left( - p_g - \rho v^2 \right)\Big|_{y=y_{\max}} \Delta x. \tag{46}$$

The control-volume extent is

$$[x_{\min}, x_{\max}] \times [y_{\min}, y_{\max}] = [-1, 2] \times [-1, 1].$$

Given the angle of attack $\alpha$ (we use $\alpha = 4°$), we rotate $(F_x, F_y)$ into the lift/drag directions:

$$L = -F_x \sin\alpha + F_y \cos\alpha, \qquad D = F_x \cos\alpha + F_y \sin\alpha. \tag{47}$$

We then non-dimensionalize using a fixed dynamic-pressure scaling $q_\infty$ (we use $q_\infty = 0.0050$) and chord length $c = 1$:

$$C_L = \frac{L}{q_\infty c}, \qquad C_D = \frac{D}{q_\infty c}. \tag{48}$$

We adopt an optimization formulation that maximizes $C_L$ while enforcing $C_D \leq C_D^{\max}$ via a hinge penalty:

$$\mathcal{L}_{\mathrm{aero}} = - C_L + \lambda_{\mathrm{cd}} \left[ \max\left( 0,\, C_D - C_D^{\max} \right) \right], \tag{49}$$

where $C_D^{\max} = 0.020$ and $\lambda_{\mathrm{cd}} = 100$.

**3D Car Surface Pressure Prediction and Optimization: DrivAerNet++.**   To evaluate model performance on complex 3D non-uniform point clouds and challenging shape optimization, we conduct experiments on the **DrivAerNet++** dataset. This dataset is a large-scale, high-fidelity CFD benchmark for automotive external aerodynamics and aerodynamic design, containing on the order of $8k$ diverse vehicle geometries (covering canonical fastback/notchback/estateback styles and multiple underbody and wheel configurations), together with high-resolution surface point clouds and surface pressure fields. For each vehicle, we use the surface point cloud coordinates as input and predict the surface pressure values. We follow the official DrivAerNet++ train/validation/test split for training and evaluation. The optimization objective is to minimize the drag coefficient of a given initial vehicle shape, while constraining the optimized geometry not to deviate excessively from the initial design (preserving key design elements). The drag coefficient is defined as

$$C_D \;=\; \frac{D}{\frac{1}{2}\rho_\infty U_\infty^2 A_{\text{ref}}}, \tag{50}$$

where $\rho_\infty$ and $U_\infty$ are the freestream density and velocity, and $A_{\text{ref}}$ is the reference area (most commonly the frontal area). The drag force $D$ is decomposed as

$$D = \underbrace{\int_{\partial\Omega_b} (-p\,\mathbf{n})\cdot\mathbf{e}_D\,\mathrm{d}S}_{D_p \text{ (pressure drag)}} + \underbrace{\int_{\partial\Omega_b} (\boldsymbol{\tau}\mathbf{n})\cdot\mathbf{e}_D\,\mathrm{d}S}_{D_\tau \text{ (skin-friction drag)}}, \tag{51}$$

where $\partial\Omega_b$ denotes the exterior surfaces of the vehicle and attachments (body, mirrors, wheels, etc.). Since the contribution of wall shear stress is quite small compared to pressure for our setting, we approximate the drag using the predicted surface pressure via a discrete surface integral:

$$D_x \;\propto\; -\sum_{i=1}^{N} p_i\, n_{x,i}\,\Delta A_i,$$

where $n_{x,i}$ is the $x$-component of the surface normal $\mathbf{n}_i$ and $\Delta A_i$ is the local area element. We then optimize the drag coefficient by minimizing this predicted drag proxy. The final optimization objective is

$$\mathcal{L}_{\text{total}} = \mathcal{L}_{\text{drag}} + \lambda_{\text{reg}}\,\|\mathbf{z} - \mathbf{z}_{\text{init}}\|_2^2,$$

with $\lambda_{\text{reg}} = 0.001$.

**Details.**   Dataset details and training configurations can be found in Tables 5 and 6 separately.

*Table 6.* Training configuration shared across all baselines.

| Benchmarks | Training Configuration (Shared in All Baselines) | | | | | |
|---|---|---|---|---|---|---|
| | **Loss** | **Epochs** | **Max LR** | **Optimizer** | **Batch Size** | **#Points** |
| Helmholtz (2D-Regular) | Relative L1 | 200 | $5 \times 10^{-4}$ | AdamW + CosineAnnealing | 32 | 4096 |
| AirFoil (2D-Irregular) | Relative L1 | 200 | $5 \times 10^{-4}$ | AdamW + CosineAnnealing | 8 | 4096 |
| DrivAerNet++ (3D-Irregular) | HuberLoss | 200 | $5 \times 10^{-4}$ | AdamW + CosineAnnealing | 4 | 50000 |

## D.2. Baselines

We compare GANO with representative learning-based PDE surrogate models and geometry-optimization pipelines, covering both strong transformer-style operators and widely used point-cloud encoders. To test GANO's performance on inverse problem and optimization, we choose four optimization models based on DEEPONET, U-NET, CORAL and FNO for 2D settings. Specifically, in these baselines, DEEPONET, U-NET and CORAL optimize Class-Shape Transformation (CST) parameters for airfoil optimization (w/o CORAL) and Fourier coefficients for Helmholtz inversion, whereas FNO directly optimizes the discretized SDF values on the grid for Helmholtz inversion. For vehicle optimization we compare PHYSGEN.

**TRANSOLVER (Wu et al., 2024a).**   TRANSOLVER is a state-of-the-art transformer surrogate for PDE field prediction on irregular point sets. It introduces a *slice–transform–deslice* mechanism that aggregates point features into a small set of slice tokens for global interaction, and then propagates the updated information back to per-point predictions. In all experiments, we train TRANSOLVER under the same data splits and supervision as our method.

**TRANSOLVER++ (Luo et al., 2025).** TRANSOLVER++ is an enhanced successor to TRANSOLVER, designed to further improve efficiency and scalability on *larger-scale geometries and higher-resolution discretizations* (i.e., PDE prediction on large point/mesh sets), and it reports stronger performance on standard PDE/geometry field prediction benchmarks. We include it as a more powerful and more scalable token/slice-based Transformer solver baseline.

**AEROGTO (Liu et al., 2025).** AEROGTO proposes a Graph-Transformer Operator tailored for *large-scale 3D vehicle aerodynamics* data: it models irregular discrete points (or surface samples from meshes) using graph/neighbor relations, and combines them with Transformer-style global interactions for aerodynamic field learning, specifically targeting large-scale vehicle aerodynamics prediction. We include it as a strong vehicle-aerodynamics baseline that couples graph operators with Transformers.

**POINTNET++ (Qi et al., 2017).** POINTNET++ is a hierarchical point-cloud encoder that learns multi-scale local features via neighborhood grouping and feature aggregation. We use POINTNET++ as a strong point-set baseline by attaching a regression head to predict the target physical fields at query points. This baseline represents standard point-cloud representation learning for high-fidelity field prediction.

**DEEPONET (Shukla et al., 2024).** Shukla et al. introduced a modular design framework that utilizes DEEPONET as a surrogate model to predict full flow fields (velocity, pressure, and density) for constrained aerodynamic shape optimization. They claimed that integrating their surrogate with the Dakota optimization toolkit reduced online optimization costs by orders of magnitude while maintaining high fidelity compared to traditional CFD solvers.

**U-NET (Rehmann et al., 2025).** Rehmann et al. proposed a Surrogate-Based Differentiable Pipeline that replaces non-differentiable CAE components with a U-NET surrogate. Implemented within the Tesseract framework, their model learns the mapping from a Signed Distance Field (SDF) to full flow fields, enabling end-to-end gradient-based aerodynamic shape optimization without the need for adjoint solvers.

**CORAL (Serrano et al., 2023).** CORAL is a coordinate-based neural-field framework for operator learning on general geometries. It encodes input and output functions with implicit neural representations and learns operators in the latent space, allowing PDE prediction across different meshes, resolutions, and irregular domains. We include CORAL as a representative INR-based operator learning baseline.

**PHYSGEN (You et al., 2025).** PHYSGEN is a physics-grounded 3D shape generation method for industrial design. It learns a shared shape-and-physics latent space with a VAE and performs physics-guided flow matching to generate physically plausible geometries. We include PhysGen as a representative physics-guided generative model baseline.

### D.3. The Details of VAE Baseline

We follow TRIPOPTIMIZER to build a triplane VAE geometry baseline. The encoder takes surface point cloud, applies point-wise positional encoding and several 1D residual blocks, then aggregates global context with an attention-based pooling module to predict $(\mu, \log \sigma^2)$ of $q_\phi(\mathbf{z} \mid X)$ and samples a latent $\mathbf{z}$ via reparameterization. The decoder maps $\mathbf{z}$ to three orthogonal TriPlanes$(F_{xy}, F_{xz}, F_{yz})$ using a $1 \times 1$ projection followed by a U-NET backbone; for 3D query point $\mathbf{x}$, it bilinearly samples features from the three planes, concatenates them, and feeds an MLP predicting voxel occupancy or SDF value. This configuration allows us to evaluate the specific impact of auto-decoder compared to the variational inference approach.

### D.4. Evaluation Metrics

We report *relative* $\ell_1$ and $\ell_2$ errors for field prediction on each test sample. Let $y \in \mathbb{R}^{N \times C}$ denote the ground-truth target (e.g., physics field values evaluated at $N$ query locations with $C$ channels), and let $\hat{y} \in \mathbb{R}^{N \times C}$ be the model prediction. We define the entry-wise error $e = \hat{y} - y$ and compute:

**Relative $\ell_1$ (Rel. L1).**

$$\text{Rel. L1}(\hat{y}, y) \triangleq \frac{\|\hat{y} - y\|_1}{\|y\|_1} = \frac{\sum_{i=1}^{N} \sum_{c=1}^{C} |\hat{y}_{i,c} - y_{i,c}|}{\sum_{i=1}^{N} \sum_{c=1}^{C} |y_{i,c}|}. \tag{52}$$

**Relative $\ell_2$ (Rel. L2).**

$$\text{Rel. L2}(\hat{y}, y) \triangleq \frac{\|\hat{y} - y\|_2}{\|y\|_2} = \frac{\left(\sum_{i=1}^{N} \sum_{c=1}^{C} (\hat{y}_{i,c} - y_{i,c})^2\right)^{1/2}}{\left(\sum_{i=1}^{N} \sum_{c=1}^{C} (y_{i,c})^2\right)^{1/2}}. \tag{53}$$

**Dataset-level reporting.** For a test set $\{(y^{(k)}, \hat{y}^{(k)})\}_{k=1}^{K}$, we report the mean of per-sample relative errors:

$$\text{Rel. L1} = \frac{1}{K} \sum_{k=1}^{K} \text{Rel. L1}\left(\hat{y}^{(k)}, y^{(k)}\right), \qquad \text{Rel. L2} = \frac{1}{K} \sum_{k=1}^{K} \text{Rel. L2}\left(\hat{y}^{(k)}, y^{(k)}\right). \tag{54}$$

For geometry representation and reconstruction, let the predicted point set be $P = \{p_i\}_{i=1}^{N}$, and the ground-truth point set be $G = \{g_j\}_{j=1}^{M}$, with Euclidean distance $\|\cdot\|_2$. For the F1-score, we use a distance threshold $\tau$ to determine whether two points are considered a match.

**F1-score.** We first define precision and recall at threshold $\tau$:

$$\text{Prec}(\tau) = \frac{1}{|P|} \sum_{p \in P} \mathbf{1} \left[ \min_{g \in G} \|p - g\|_2 \leq \tau \right], \tag{55}$$

$$\text{Rec}(\tau) = \frac{1}{|G|} \sum_{g \in G} \mathbf{1} \left[ \min_{p \in P} \|g - p\|_2 \leq \tau \right]. \tag{56}$$

The F1-score is the harmonic mean of precision and recall:

$$\text{F1}(\tau) = \frac{2 \, \text{Prec}(\tau) \, \text{Rec}(\tau)}{\text{Prec}(\tau) + \text{Rec}(\tau)}. \tag{57}$$

**Chamfer Distance.** The symmetric Chamfer Distance between $P$ and $G$ is defined as

$$\text{CDist}(P, G) = \frac{1}{|P|} \sum_{p \in P} \min_{g \in G} \|p - g\|_2^2 + \frac{1}{|G|} \sum_{g \in G} \min_{p \in P} \|g - p\|_2^2. \tag{58}$$

**Earth Mover's Distance.** When $|P| = |G| = N$ and a one-to-one correspondence is assumed, EMD is the minimum matching cost:

$$\text{EMD}(P, G) = \min_{\phi: P \to G} \frac{1}{N} \sum_{p \in P} \|p - \phi(p)\|_2, \tag{59}$$

where $\phi$ is a bijection (i.e., a permutation-based assignment) that minimizes the total transport cost.

**High-fidelity CFD verification for $C_L/C_D$ (AirFoil).** To accurately evaluate the aerodynamic performance of the optimized airfoil geometries, we perform high-fidelity CFD verification using COMSOL. Following the same setting as AirFoil_9k, the CFD simulations are performed at a freestream Mach number of 0.1, Reynolds number of 9M, and angles of attack 4 degrees, using the $k - \omega$ SST turbulence model. Lift and drag are then computed via a control-surface integral on a circular far-field control surface with radius $R = 300\,\text{m}$ and the net aerodynamic force is obtained the net aerodynamic force by integrating the traction over this boundary, followed by standard non-dimensionalization to report $C_L$ and $C_D$.

**High-fidelity CFD verification for $C_D$ (Vehicle).** To evaluate the aerodynamic performance of our optimized vehicle geometries, we perform high fidelity CFD simulations using OpenFOAM to compute the drag coefficient. A uniform inlet freestream velocity of 30 m/s, aligned with the vehicle's longitudinal axis and directed toward the frontal surface, is prescribed to simulate standard automotive aerodynamic operating conditions. We employ the steady-state SimpleFOAM solver together with the $k - \omega$ SST turbulence model. Each simulation proceeds through 1500 iterations to ensure convergence. The final drag coefficient is obtained by averaging the result throughout the last 500 iterations.

# E. Additional Experiments

### E.1. Slice visualization

TRANSOLVER assigns input points into multiple slices, where each slice is intended to represent a latent physical state. Points with similar activation degree within a slice should therefore share similar latent physical conditions. As shown in Fig. 11, we visualize the slices to compare our GI-TRANSOLVER against the original TRANSOLVER.

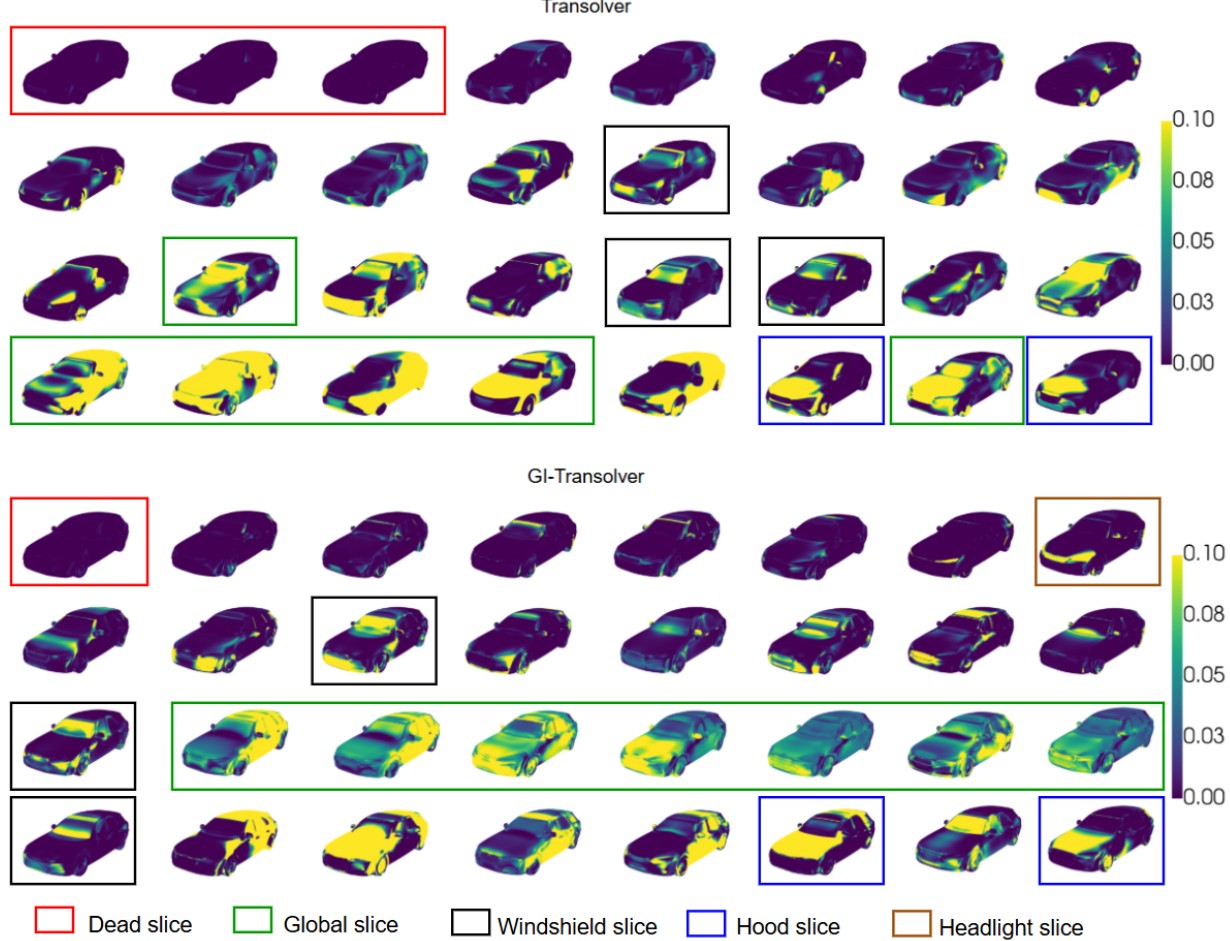

*Figure 11.* **Comparison of slices of GI-Transolver and Transolver.** 1) Red indicates dead slices. A dead slice refers to a slice with very low activation values and no clear semantic meaning. GI-Transolver has fewer dead slices than Transolver. 2) Green indicates global slices. A global slice has relatively high activation values over the whole object. Compared with localized activations, such globally high activations can be interpreted as corresponding to a global physical feature. Due to our colorbar setting, the visual differences within the range from 0.03 to 0.08 are not very pronounced, so the slice may appear blurry for GI-Transolver. 3) Black indicates the windshield slice. Both models capture this region. 4) Blue indicates the hood slice. The hood region learned by our model is more complete than that learned by Transolver. 5) Brown indicates the headlight slice. In contrast to ours, Transolver shows a less localized pattern on this region.

### E.2. The Effects of Gaussian Noise

To study whether latent perturbation degrades reconstruction fidelity, we inject different scale isotropic Gaussian noise into the latent code during training and evaluate reconstruction quality without noise injection on unseen geometry. This experiment is conducted on DrivAerNet++. Specifically, we train DEEPSDF auto-decoders with noise standard deviation $\sigma \in \{0, 0.005, 0.01, 0.05\}$ and report F1-score, Chamfer Distance (CDist), and Earth Mover's Distance (EMD). As shown in Table 7, adding Gaussian noise does *not* noticeably reduce reconstruction quality: F1-score remains stable around 0.911-0.912, while CDist and EMD show negligible changes across all settings. This suggests that, owing to the strong representational capacity of DEEPSDF, moderate latent noise injection can be absorbed without sacrificing geometric

fidelity.

*Table 7.* Ablation of noise scale.

| $\sigma$ | 0 | 0.005 | 0.01 | 0.05 |
|---|---|---|---|---|
| F1-score ↑ | 0.911 | 0.912 | 0.912 | 0.911 |
| CDist ↓ | 8.26e-05 | 8.25e-05 | 8.24e-05 | 8.26e-05 |
| EMD ↓ | 6.17e-02 | 6.11e-02 | 6.23e-02 | 6.18e-02 |

### E.3. The Effects of Number of Sensors

We further investigate how the number of boundary sensors affects inverse scattering reconstruction quality. We vary the sensor count from extremely sparse measurements (10 sensors) to dense observations (400 sensors), while keeping all other settings fixed. Fig. 12 shows representative reconstructions and the corresponding pixel-wise error maps. Overall, our method remains robust even under highly sparse sensing. With only 10 sensors, the recovered obstacle already captures the correct global silhouette and preserves the main concave/convex boundary structures. Increasing the number of sensors provides only marginal improvements: the reconstruction errors fluctuate within a narrow range (approximately 6%-7% across all sensor counts in this example), and the predicted shapes remain visually similar. These results indicate that our latent-space inversion is not overly reliant on dense measurements.

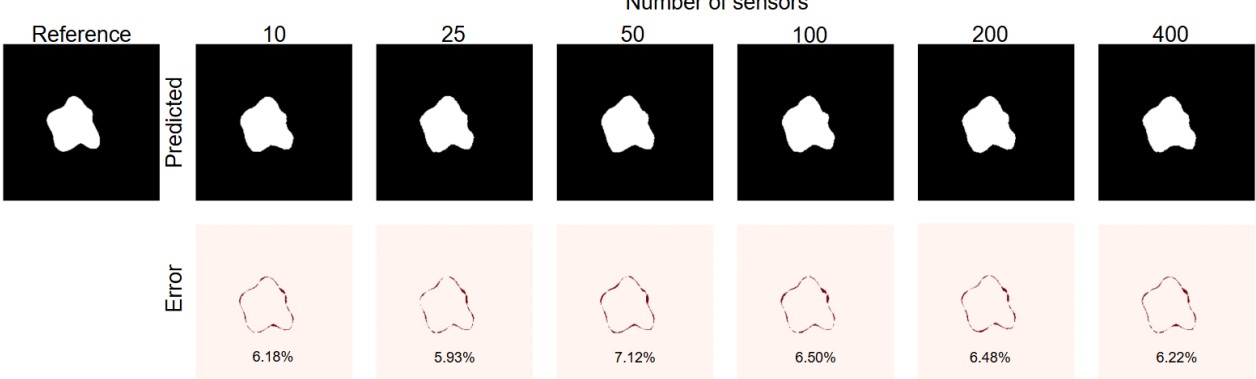

*Figure 12.* Ablation of the number of sensors on Helmholtz inversion.

### E.4. The Effects of Number of Sampling Points for STABLESDF

We study how the number of input samples at inference time affects reconstruction quality of STABLESDF on DrivAerNet++. While our model is trained with 50000 surface points per shape, we evaluate reconstruction metrics using different numbers of sampled surface points, i.e., 10000, 20000, and 50000, and report F1-score, Chamfer Distance (CDist), and Earth Mover's Distance (EMD). As shown in Table 8, reducing the number of samples significantly degrades reconstruction quality. In particular, using only 10000 points leads to a clear drop in F1-score (0.693 vs. 0.944) and a substantially higher CDist, indicating that sparse sampling struggles to preserve fine details and structures. Increasing the input to 20000 points markedly improves reconstruction quality (F1-score 0.854; CDist $10.3 \times 10^{-5}$), yielding results that are already acceptable. As expected, performance further improves and saturates when matching the training-time setting (50000 points). Overall, these results suggest that while STABLESDF benefits from dense sampling for high-fidelity vehicle reconstruction, a moderate input resolution (around 20000 points) can provide a practical accuracy-efficiency trade-off at inference time.

*Table 8.* Ablation of sampling points numbers of STABLESDF.

| Points | 10000 | 20000 | 50000 |
|---|---|---|---|
| F1-score ↑ | 0.693 | 0.854 | 0.944 |
| CDist ↓ | 16.2e-05 | 10.3e-05 | 6.62e-05 |
| EMD ↓ | 6.15e-02 | 6.28e-02 | 6.20e-02 |

### E.5. Ablation Study of Geometric Injection

We study different strategies for injecting geometric latent $\mathbf{z}$ into TRANSOLVER: (i) **Additive injection**, directly adding $\mathbf{z}$ to slice tokens; (ii) **Cross-attention injection**, generating geometry embeddings from $\mathbf{z}$ through MLP and interacting with slices via cross-attention; (iii) **Gated injection**, using slice-conditioned gates to selectively inject $\mathbf{z}$. As shown in Table 9, on DrivAerNet++, gated injection provides the best performance, outperforming simpler additive fusion and cross-attention.

*Table 9.* Ablation of geometric injection methods. Additive Injection (Add), Cross-attention Injection (CA), and the proposed Gated Injection (GANO - ours).

| Method | Add | CA | GANO (ours) |
|---|---|---|---|
| **Rel. L2** ↓ | 18.2% | 18.5% | 17.8% |
| **Rel. L1** ↓ | 16.8% | 17.3% | 16.5% |

### E.6. The Effects of Remeshing-free Projection

We employ a remeshing-free projection scheme that maps query points onto the zero-level set of the updated surface to replace time-consuming remeshing. We evaluate the performance of this projection in Fig. 13. As shown in the histogram in Fig. 13**a**, we add a Gaussian perturbation to surface points, the resulting distribution (red) exhibits significant deviation from the surface. After applying our method, the distances converge effectively to zero (green), demonstrating that the points now lie on the surface. This is visually corroborated in Fig. 13**b**, where the point cloud, initially colored by distance errors, is corrected to perfectly align with the car geometry (shown in black). Notably, this operation is computationally efficient, projecting $10^4$ points in approximately 30 ms within only 5 iterations. Table 10 further evaluates projection success rates under different absolute SDF thresholds on a larger experiment scale. Fig. 14 demonstrates that the Jacobian norm of the StableSDF output with respect to $\mathbf{x}$ remains around 1 before and after perturbation, which is a desirable property for an SDF field.

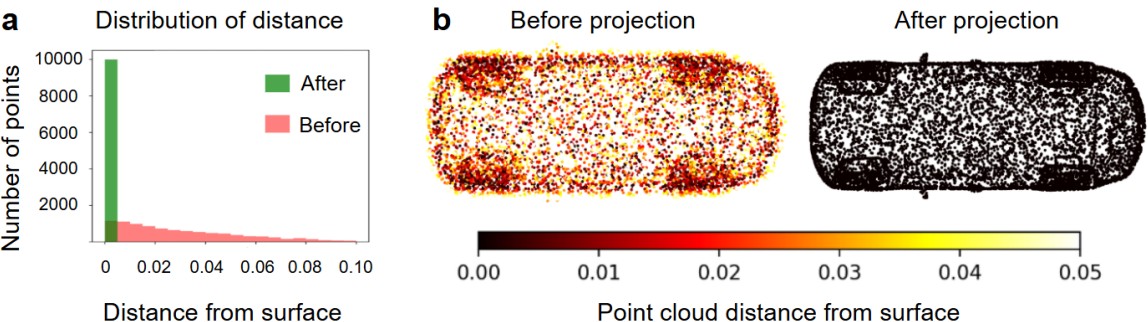

*Figure 13.* **Remeshing-free projection for boundary-consistent queries. a**. Histogram of point-to-surface distances before (red) and after (green) projection. **b**. Point set colored by distance to the surface before (after) projection. We project $10^4$ points onto the updated surface in $\approx$ 30 ms within 5 iterations.

### E.7. Computational Cost

**Forward prediction.** We report inference efficiency in Table 11 in terms of per-sample latency, peak GPU memory, and parameter count. Overall, our method shows similar speed and memory use as TRANSOLVER: on Helmholtz we match its memory footprint (5.83 vs. 5.82 GB) while being faster (4.97 vs. 5.89 ms), and on AirFoil_9k we achieve comparable

*Table 10.* **Projection success rates under different absolute SDF thresholds.** The statistics are computed on 100 vehicles, each with $10^5$ query points, resulting in $10^7$ points in total. A point is considered successfully projected if its absolute SDF value is below the threshold.

| Threshold | Before projection | After projection |
|-----------|-------------------|------------------|
| $10^{-6}$ | 0.0024% | 98.6350% |
| $10^{-5}$ | 0.0231% | 99.7322% |
| $10^{-4}$ | 0.2281% | 99.9739% |
| $10^{-3}$ | 2.2684% | 99.9998% |

**a.**

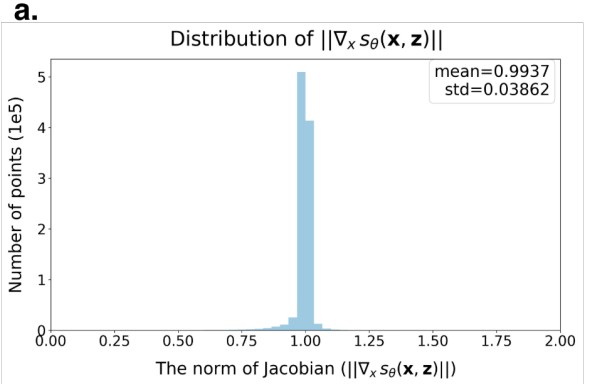

**b.**

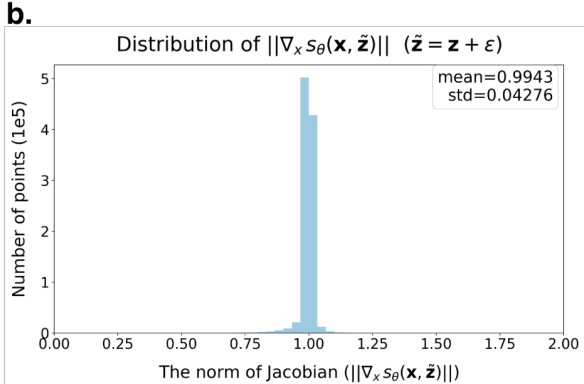

*Figure 14.* **Distribution of the Jacobian norm of the StableSDF output with respect to x: a.** before random perturbation and **b.** after random perturbation. The statistics are computed from 100 randomly sampled test vehicles, with 10,000 points sampled from each vehicle.

latency (4.53 vs. 4.63 ms) with fewer parameters (2.49M vs. 3.78M), at a modest increase in memory (2.40 vs. 1.68 GB). In contrast, AEROGTO is consistently a high-memory outlier (44.04/7.79/74.00 GB on Helmholtz/AirFoil_9k/DrivAerNet++) which significantly limits its training speed and scalability to dense mesh, whereas our method stays in the single-digit GB regime and is also faster.

**Optimization.** We report the end-to-end runtime of the full pipeline for optimization/inversion of Helmholtz, airfoil, and vehicle, covering geometry encoding, forward prediction, points projection time, and full round time. All experiments are conducted on a single NVIDIA A100 GPU. We run the optimizer for 100 steps, using 100 observed points on Helmholtz; for 100 steps, using about 50k mesh points on airfoil; for 40 steps, using about 50k surface points on a car. Summarized in Table 12, results demonstrate the efficiency of our pipeline.

*Table 11.* Performance comparison on Helmholtz, Airfoil, Vehicle datasets. We report the **Inference Time per sample** (ms), **Training GPU Memory** (MB), and Parameters (M). Note that the GPU memory consumption is measured during training and depends on the Batch Size (BS). The specific settings for training BS and number of points per sample ($N$) are: **Helmholtz** (BS=32, $N$=4096), **Airfoil** (BS=8, $N$=4096), and **Vehicle** (BS=4, $N$=50k).

| Model | Helmholtz | | | Airfoil | | | Vehicle | | |
|---|---|---|---|---|---|---|---|---|---|
| | Time (ms) | Mem. (MB) | Param. (M) | Time (ms) | Mem. (MB) | Param. (M) | Time (ms) | Mem. (MB) | Param. (M) |
| AEROGTO | 7.43 | 45102 | 2.09 | 11.54 | 7979 | 2.09 | 66.16 | 75780 | 2.08 |
| DEEPONET | 0.47 | 441 | 0.11 | 0.31 | 179 | 0.13 | - | - | - |
| FNO | 3.12 | 677 | 4.74 | - | - | - | - | - | - |
| POINTNET++ | 223.00 | 4503 | 0.94 | 299.47 | 1150 | 0.96 | 255.97 | 11502 | 1.03 |
| TRANSOLVER | 5.89 | 5957 | 1.73 | 4.63 | 1725 | 3.77 | 9.05 | 9218 | 2.00 |
| TRANSOLVER++ | 4.88 | 7807 | 1.99 | 5.05 | 1015 | 2.13 | 11.14 | 6444 | 1.74 |
| U-NET | 18.55 | 15313 | 4.24 | 15.52 | 3667 | 4.24 | - | - | - |
| GANO (ours) | 4.97 | 5972 | 1.83 | 4.53 | 2456 | 2.48 | 32.08 | 9310 | 2.07 |

*Table 12.* Runtime breakdown (in seconds) of the optimization/inversion pipeline on Helmholtz, airfoil, vehicle, where "-" means not required.

| Time (s) | Helmholtz | Airfoil | Vehicle |
|---|---|---|---|
| Geometry encoding | - | 0.53 | 1.10 |
| Forward prediction | 1.92 | 1.19 | 0.64 |
| Points projection | - | 16.35 | 4.10 |
| Full round | 3.93 | 35.69 | 7.39 |

# F. Additional Results

We provide additional qualitative visualizations of the experimental results here. Fig. 15 shows the full qualitative comparison for the baseline forward problem in the Helmholtz experiment. In (a), the $45°$ incidence angle is out-of-distribution, while in (b), the $0°$ incidence angle is seen during training. For each case, we visualize the real part and imaginary part of the scattered field, as well as the corresponding error maps. Fig. 16 shows the full qualitative comparison for the baseline forward problem in the airfoil experiment. We report the predicted fields and error maps for the horizontal velocity, vertical velocity, and pressure. Fig. 17 illustrates the evolution of the airfoil shape and the automatically projected mesh during the GANO-based airfoil optimization process. Fig. 18 gives a comparison of geometry-informed UNET and DEEPONET, CORAL, and GANO on 2D Helmholtz task. Fig. 19 compares the results after optimizing the same Fastback and Estateback car for 20 steps using STABLESDF and DEEPSDF under the same settings. Fig. 20 visualizes the results of GANO and PHYSGEN in predicting the pressure field.

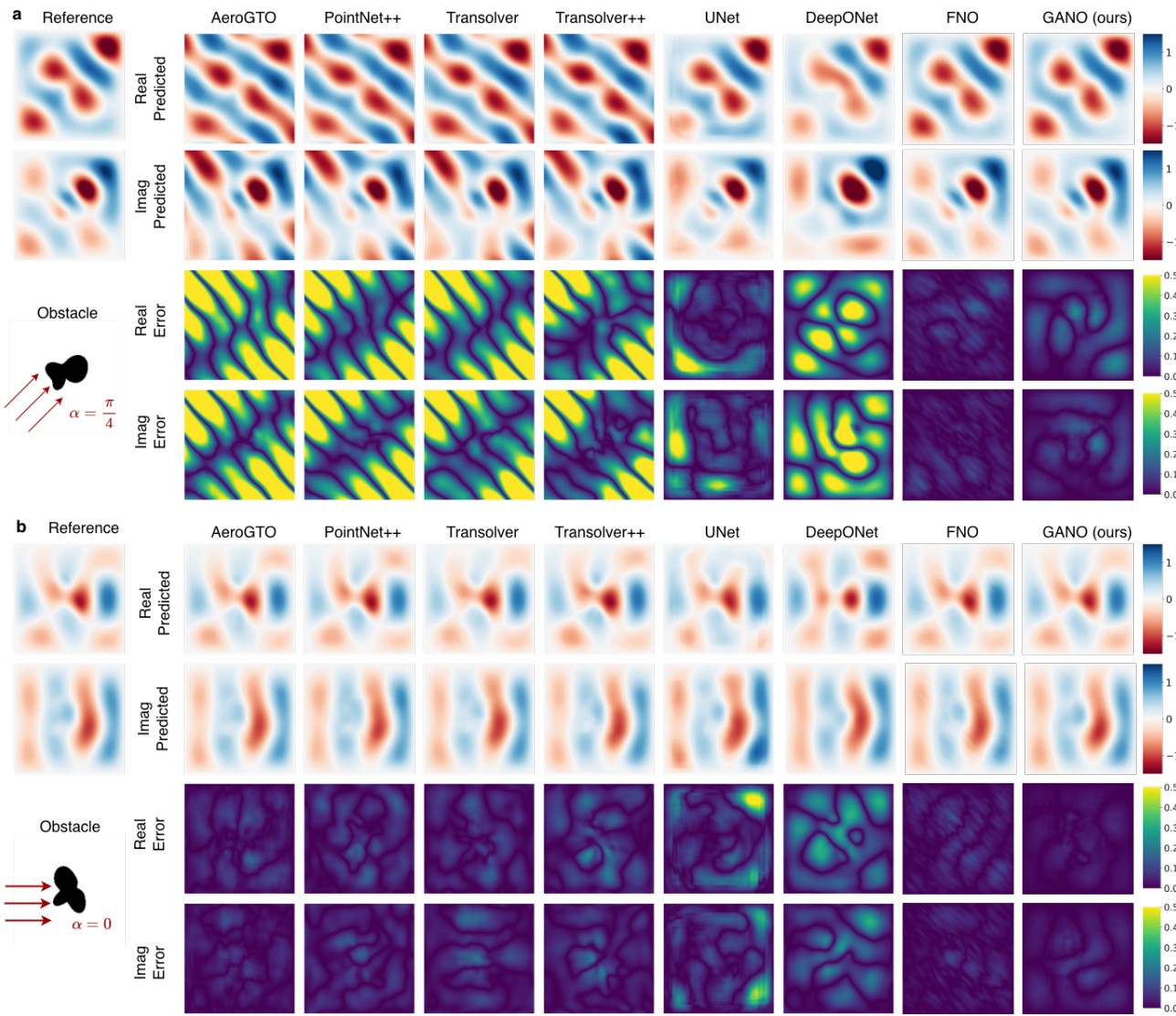

*Figure 15.* Full results on Helmholtz dataset.

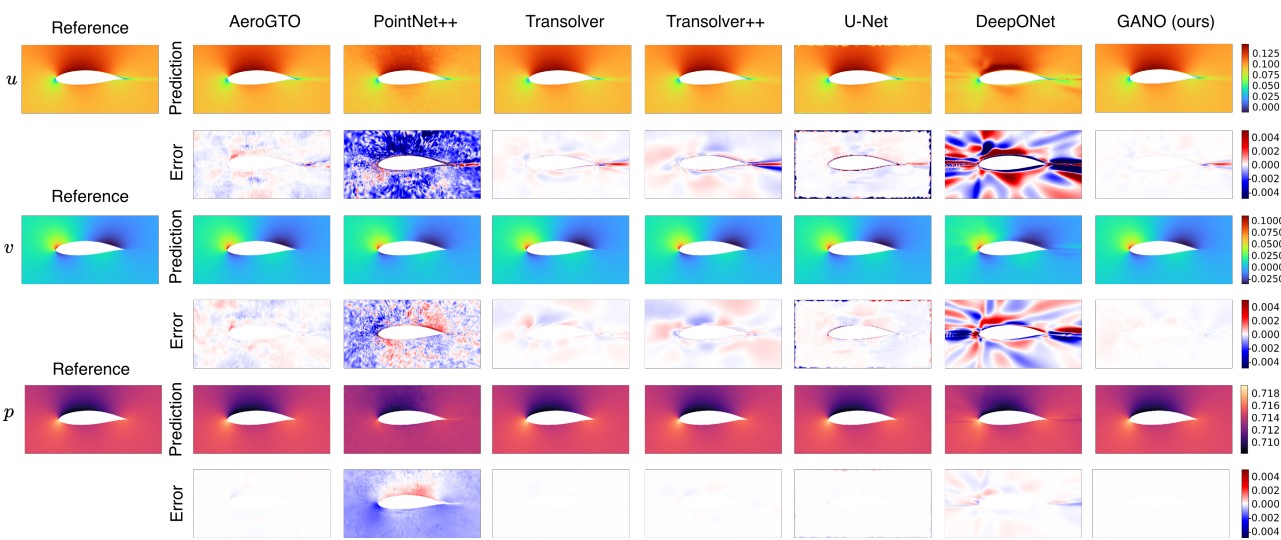

*Figure 16.* Full results on Airfoil dataset.

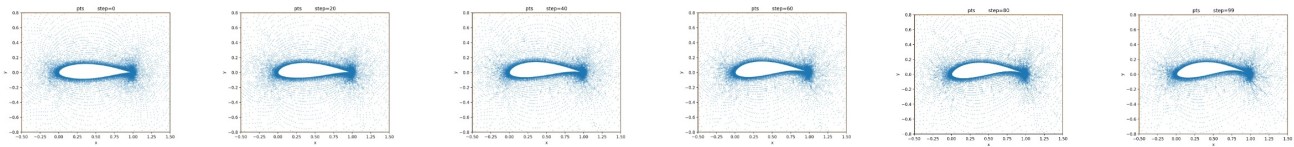

*Figure 17.* Remeshing free optimization process for an airfoil, showing how sampling points changes with the geometry.

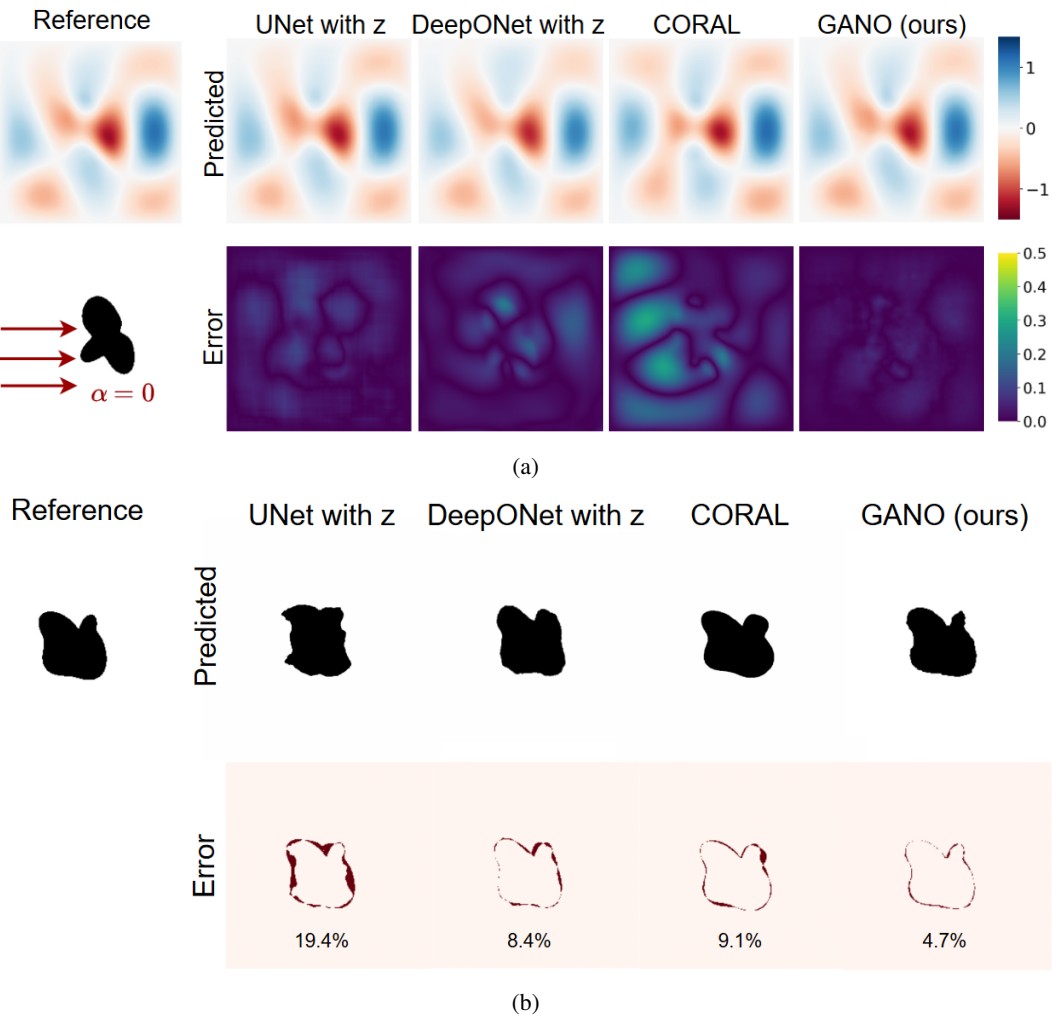

*Figure 18.* **2D Helmholtz tasks: comparison of geometry-informed UNet and DeepONet, CORAL, and GANO. a.** Forward task. In CORAL, Fourier shape parameters are first used to generate the obstacle boundary, and the region between this boundary and a fixed outer boundary is then interpolated into a 2-channel coordinate field as the geometric input encoded by CORAL. When predicting the latent physical field, the geometric latent and the incident angle are concatenated as the input to the MLP. Other settings are consistent with its open-source code. **b.** Inversion task.

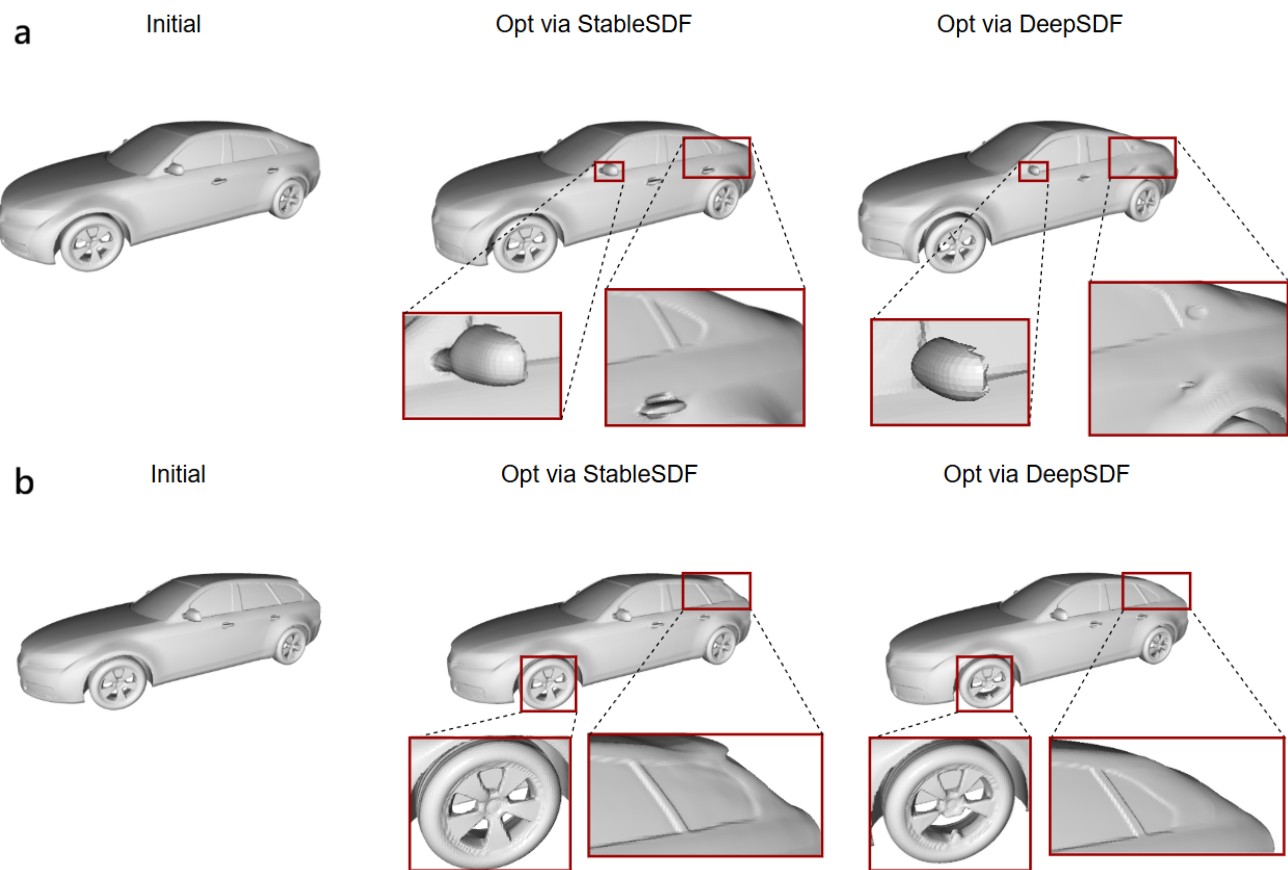

*Figure 19.* **Comparison of the results after optimizing the same Fastback and Estateback car for 20 steps using StableSDF and DeepSDF under the same settings. a.** After optimization with DeepSDF, the deformation of the car details is relatively severe: the side mirrors detach from the car body, and the rear door handle as well as the rear window also undergo severe deformation. **b.** After optimization with DeepSDF, the wheels show obvious damage, and the rear back of the car deviates severely from the original design, changing from an Estateback into a Fastback.

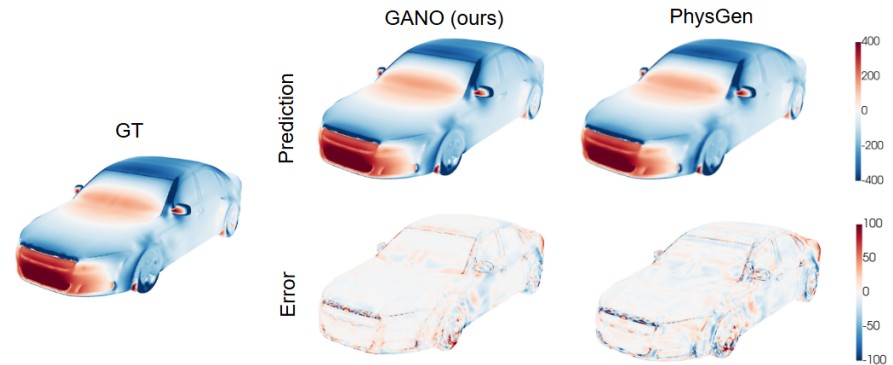

*Figure 20.* **Comparison of the results of GANO and PhysGen in predicting the pressure field.**

