# OpenReview forum: "Geometry-Aware Neural Optimizer for Shape Optimization and Inversion"
_ICML.cc/2026/Conference — ICML 2026 regular_

### Official Review · Reviewer_VNRW · 2026-03-05

**Soundness:** 3
**Presentation:** 4
**Significance:** 3
**Originality:** 3
**Overall Recommendation:** 4
**Confidence:** 4

**Summary:**

This paper presented an end-to-end differentiable framework that unifies geometry representation, field-level prediction, and automated shape optimization/inversion in a single latent-space loop, named as  GANO, which  enables efficient, remeshing-free geometry updates with part-wise control.

**Compliance With Llm Reviewing Policy:**

Affirmed.

**Key Questions For Authors:**

No.

**Limitations:**

The limitations are well discussed, the potential negative societal impact are zero.

**Strengths And Weaknesses:**

Soundness:Totally, the work of this paper is technically sound.  Theoretical analysis or experimental results are well presented. The experiments are well-designed. The strength and a weakness aie well discussed.

Presentation: The submission clearly written and well structured.

Significance:The paper address an  relevant problem. It advances the  practice in machine learning.

Originality: This work introduce new methods and  perspectives that advance the relevant field.

---

> ### Author Rebuttal · Authors · 2026-03-28
>
> Thank you for your time and valuable feedback, as well as your positive comments and interest in our paper!

---

### Official Review · Reviewer_r7Qz · 2026-03-14

**Soundness:** 3
**Presentation:** 3
**Significance:** 3
**Originality:** 2
**Overall Recommendation:** 4
**Confidence:** 1

**Summary:**

This paper proposes GANO, a framework that combines a latent geometry representation based on DeepSDF, a geometry-injected forward surrogate build on Transolver, and a latent-space optimization loop with remeshing-free projection and optional part-wise control via null-space projection. The paper evaluates the approach on three settings: 2D Helmholtz inverse scattering, 2D airfoil flow prediction and optimization, and 3D vehicle pressure prediction and drag reduction.  The article is well-organized and well-written, and the experiments are fairly comprehensive. However, it needs further clarification on more detailed issues, as seen in the content below.

**Compliance With Llm Reviewing Policy:**

Affirmed.

**Final Justification:**

Overall, this paper is reasonably complete, and I still recommend a weak accept.

**Key Questions For Authors:**

1. Can the authors more clearly isolate the benefit of STABLESDF relative to a standard DeepSDF baseline during optimization, not only interpolation?
2. Does STABLESDF use any explicit eikonal-style or other regularization to enforce that the learned field behaves like a true signed distance field, especially for the projection step in Eq. (10)? If not, why is the projection still reliable in practice?
3. To justify the necessity of the chosen geometry representation, can the authors comment on why only a VAE/occupancy comparison was included, and whether more modern geometry models were considered? The current comparison does not fully establish that an auto-decoder SDF formulation is the most compelling design choice among current advanced alternatives.

**Limitations:**

The limitations discussion is partially adequate but could be improved. In the current benchmarks, the shapes within each domain appear fairly similar in overall structure (e.g., airfoils, vehicles within a constrained design family), which likely lowers the difficulty of the geometry modeling problem. It remains unclear how well the proposed geometry representation and optimization loop would scale to settings with substantially richer, more heterogeneous shape families. This applicability regime should be discussed more explicitly.

**Strengths And Weaknesses:**

# Strengths
- The overall pipeline is technically reasonable, and the paper includes a substantial set of experiments across multiple tasks and dataset.
- I also appreciate that the paper includes some theoretical discussion for the denoising mechanism and the projection step.
- The paper is clearly written and well organized
- While I do not view the method as highly original, the paper does provide a reasonable systems-level integration of geometry representation, forward prediction, and optimization.

# Weakness
My main concern is that the overall framework appears to be largely a combination of a latent 3d modeling based on DEEPSDF and PDE surrogate prediction based on Transolver, with incremental modifications to each. The paper does make engineering and methodological improvements on top of these components, but the conceptual novelty seems moderate rather than strong.
- A major issue analyzed by the manuscript is the claim that denoising-based latent perturbation leads to smoother, more controllable geometry updates. However, the evidence shown in Fig. 3 is not fully persuasive to me. The interpolation is performed between two already similar car shapes, and only a small number of interpolation points are visualized. In such a setting, even a standard DeepSDF latent interpolation might look natural. As a result, the current figure does not clearly establish that STABLESDF provides a materially better latent geometry space than a strong DeepSDF baseline.
- The projection step is reasonable if the learned implicit field is a sufficiently accurate and well-behaved signed distance field near the surface. But the paper does not clearly state whether training includes explicit constraints such as an eikonal loss or other SDF regularization enforcing near-unit gradient norm. This matters because Eq. (10) is justified geometrically when the field behaves like a proper SDF locally. Even if such regularization is used during training, it is also unclear whether the decoded field at newly updated latent codes continues to satisfy the necessary local properties during optimization. The appendix provides one projection example, but this is not enough to establish robustness.
- Appendix E.6 gives one example where points return to the zero level set after projection, but several details remain unclear: whether this example is from the training or test set, how often projection succeeds across a large set of unseen shapes, and what “successful” projection means quantitatively. I would like to see broader statistics over many test instances.
- The author should add comparison against the work, PhysGen, which is also highly similar to the motivation of this work.
- Table 7 in Appendix E.2 varies the noise scale, but the reported numbers are extremely similar, making it hard to extract a strong empirical conclusion about the practical effect of noise injection on reconstruction quality. The paper argues that denoising improves stability, but the ablation as currently presented does not strongly reveal a trend.

---

> ### Author Rebuttal · Authors · 2026-03-30
>
> Thanks for your valuable comments! **Additional figures are in anonymous link: https://github.com/GANOICML/GANO_ICML2026.**
>
> **W0: Conceptual novelty**
>
> **Reply:** Our main contribution isn't a new geometry representation or PDE solver in isolation but a **differentiable, controllable, and stable optimization** frame. All modifications to existing models are introduced specifically for this purpose. StableSDF provides a stable optimization space, and GI-Transolver establishes a direct differentiable path from the physical field to geometric representation. Without this design, Transolver can only accelerate the forward problem, but optimization would require heavy sensitivity analysis. Thus, these changes are in fact essential for differentiable, stable, and controllable optimization, with theoretical and empirical support. We also introduce new techniques to support controllable optimization and improve efficiency, e.g., GANO can preserve key structures unchanged which previous models lack.
>
> **W1 & Q1: StableSDF vs DeepSDF**
>
> **Reply:** We compare them as follows:
> - Fig. 8a compares their Jacobian norms. StableSDF has a significantly smaller norm, indicating more stable deformation, consistent with Thm. A.1.
> - Fig. 8b shows robustness under latent noise perturbation. Even with large noise, StableSDF better preserves fine details.
> - Following your suggestion, we compare their optimization results. Shapes optimized with StableSDF show more stability than those using DeepSDF (Fig. 3 in the link).
>
> **W2 & Q2: Eikonal loss**
>
> **Reply:** StableSDF doesn't add Eikonal loss, consistent with DeepSDF. Yet Eq. 10 doesn't rely on the exact Eikonal property.
> - The theory of Eq. 10 relies on **regular spatial gradient** $(\|\nabla_xs(x)\|\ge m)$ rather than unit gradient norm. This is because our analysis is based on a Gauss-Newton view of the projection. Since GI-Transolver operates on a surface point cloud, the key is to reliably **reproject the points onto the updated surface**, which is supported by our theorem.
> - Empirically, the spatial gradient regularity of StableSDF is well maintained. We compute gradient norm near the surfaces of 100 test vehicles, before and after latent perturbation, using 10000 points per vehicle. In both cases, the mean norm remains ~**0.99** with a std of 0.04 (Fig. 6 in the link), supporting the regular spatial gradient condition holds after the latent code's update.
>
> **W3: Projection process**
>
> **Reply:** The paper reports projection result of a randomly selected **test** vehicle. During rebuttal, we conducted 100 experiments by randomly perturbing the surface point cloud. Before projection, the mean absolute SDF value is 0.04 (with the vehicle normalized to [-1,1]). After 5 projection steps, the mean SDF magnitude decreases to ~$10^{-7}$, the maximum SDF value below $10^{-3}$. Note that each geometry update is small during optimization. Quantitatively, we add projection success rates in Table 1 in the link.
>
> **W4: Comparison with PhysGen**
>
> **Reply:** By ICML deadline, we surveyed relevant 3D vehicle design works and compared them in Table 4, but none released code. During rebuttal, we found PhysGen released its code on **March 20**, so we followed its settings to compare on vehicles. PhysGen requires ~6 days to train (we use their released model), while GANO requires 14h. In optimization task, PhysGen takes ~240s per case, while GANO takes ~10s. Using its released model, PhysGen achieves Rel.L1/L2=0.177/0.194 (0.166/0.178 for GANO). For optimization, PhysGen is comparable to or worse than GANO, and we observe some less meaningful deformations: the side mirrors rotate inward to vehicle body and get smaller. In contrast, GANO restricts mirror deformation to better satisfy practical design constraints (Fig. 4-5 in the link).
>
> **W5: Claim in App. E.2**
>
> **Reply:** In App. E.2, our intention was to show that moderate noise does not harm reconstruction accuracy, rather than improve stability. We will revise it.
>
> **Q3: Geometry representations**
>
> **Reply:** We surveyed geometry encodings used in recent 3D vehicle optimization works and found they mainly adopt **VAE** or **occupancy** representations. Thus, we focus on these methods. In our paper, geometry representation is chosen to support the downstream optimization loop. We adopt SDF-based representation because of its **compact latent codes** and **remeshing-free projection**, which are suitable for stable and efficient geometry updates. We agree that exploring other methods is valuable, but properly integrating them into our end-to-end optimization frame and verifying whether they preserve the same efficiency may be beyond the scope of this paper. We will highlight it as future work.
>
> **Richness of dataset**
>
> **Reply:** To our knowledge, DrivAerNet++ is the largest open-source dataset for vehicle shape optimization (Fig. 2 in the link). Also, we agree that current data doesn't cover full diversity cases and will discuss it in revision.

---

> > ### Author Rebuttal · Reviewer_r7Qz · 2026-04-02
> >
> > The author has generally addressed my questions. However, the discussions on certain points, such as the experimental analysis of SDF and the final choice of geometric representation, still lack sufficient thoroughness. Overall, the paper is well-developed, and I decide to maintain my original choice.

---

> > > ### Author Response · Authors · 2026-04-03
> > >
> > > Thank you very much for your support and for maintaining your positive assessment. We also appreciate your comment that the experimental analysis of StableSDF and the discussion of geometric representation could be further strengthened. For the analysis of StableSDF, we have summarized our experiments below for your reference:
> > >
> > > 1. Comparison of StableSDF and DeepSDF in terms of the Jacobian norm with respect to $z$ (Fig. 8a).
> > >
> > > 2. Robustness comparison between StableSDF and DeepSDF under latent perturbations of $z$ (Fig. 8b).
> > >
> > > 3. We compare the effect of different training noise scales on the final shape reconstruction quality (Table 7).
> > >
> > > 4. Following your guidance, we compare the vehicle optimization results of StableSDF and DeepSDF (W1 & Q1), add statistics of the SDF gradient magnitude before and after projection to support Eq. 10 (W2 & Q2), and add projection success-rate statistics on 100 test vehicles (W3).
> > >
> > > In summary, we sincerely thank you for your insightful comments, which have helped strengthen our paper, especially the part on geometry representation. We will make sure that your suggestions are fully reflected in the final version of the paper, and we will carefully consider your insightful advice in future work.

---

### Official Review · Reviewer_q8ZS · 2026-03-14

**Soundness:** 3
**Presentation:** 3
**Significance:** 2
**Originality:** 3
**Overall Recommendation:** 4
**Confidence:** 3

**Summary:**

This paper introduces GANO, a new framework for shape optimization and inverse problems for static PDEs. It first encode geometries using DeepSFD and then makes use of a Transolver architecture to predict the associated physical field. This is coupled with a dedicated optimization loop to optimize the geometry. The paper is structured with a technical description of the algorithm, some experiments and theoretical analysis.

**Compliance With Llm Reviewing Policy:**

Affirmed.

**Final Justification:**

The rebuttal has addressed my main concerns and questions regarding the paper with new experiments and explanations.
The method, while building upon well-know method (Transolver, auto-decoding and deepsdf), explores a specific task and demonstrates strong experimental results, so I raised my score to 4.

**Key Questions For Authors:**

## Questions
- Can you provide some training metrics such as training times? I think this method requires 2 training stages making it possibly costly to train?
- Why not comparing to more specific and relevant models of the literature ? DeepONet, UNet and FNO are not the SOTA models in the field. For instance, PhysGEN, 3DID that are mentioned and solve the same problems for instance ?
- Have you tested the effect of the geometry injection in the model? How does it behave without this information? My point here would be to evaluate the benefits of your architectural choices wrt others. On the other hand, how would perform other baseline with the geometry injection (line 365-369) ?
- I couldn't find the implementation details on FNO, how is this baseline evaluated?
- Have you considered other INR methods to represent the geometries? For instance, CORAL [1]  is another work that uses implicit representations for dynamical forecasting and explains how to solve design tasks (it is based on SIRENs). Could you further detail the algorithmic differences between CORAL and GANO ?

## Minor
- Some files in the supplementary material seemed to be empty (diff_visualizationF.html, diff_visualizationE.html)

## References
[1] Operator Learning with Neural Fields: Tackling PDEs on General Geometries, Louis Serrano, Lise Le Boudec, Armand Kassaï Koupaï, Thomas X Wang, Yuan Yin, Jean-Noël Vittaut, Patrick Gallinari

**Limitations:**

Yes

**Strengths And Weaknesses:**

## Strenghts :
- The paper is overall well written and pleasant to read.
- There is a detailed theoretical analysis in the appendices
- The experimental part provides several ablations on the different model components with illustrations on their effects

## Weaknesses :
- In my understanding the method requires several training parts: one for the Stable SDF decoder and one for the neural operator model. I think a study on the training cost is missing as it could be a big bottleneck of the model. I may have misunderstood this part but it is unclear for me the differences between training/inference procedures. This also makes hard for me to identify the contribution of the work wrt to existing frameworks.
- The baselines used for the surrogate-based optimization evaluation could be updated (as mentioned in the related work and Table 4 comparison for instance).
- I think a part of the literature on implicit modeling is missing. A few word could help motivate the choice and modification of StableSDF.

---

> ### Author Rebuttal · Authors · 2026-03-30
>
> Thanks for the helpful comments! **Additional figures are in the anonymous link: https://github.com/GANOICML/GANO_ICML2026.**
>
> **W1 & Q1: Train and optimization**
>
> **Reply:** For training, we first train StableSDF and then GI-Transolver. For optimization, we freeze models and update the latent code $z$ end-to-end. Since StableSDF is an auto-decoder, we first solve a small optimization problem to infer the initial latent code $z_0$ from the initial geometry, which takes ~1s (App. E.7).
>
> For the training cost on 3D vehicles, training StableSDF on one A100 takes ~**8h**, and training GI-Transolver takes ~**6h**, resulting in 14h in total. Yet, methods like PhysGen/3DID also require multi-stage training. Since they require training diffusion models, their cost is on the order of days. For example, PhysGen takes 6.2 days, with the breakdown training costs listed as follows:
>
> Modules|Shape encoder|Pressure decoder|Drag decoder|Joint fine-tune|Diffusion|Total (PhysGen)
> |-|-|-|-|-|-|-|
> Time|24h|21h|24h|15h|64h|148h
>
> Note that the key practical metric is the cost of the entire process of **geometry optimization** (defined as inference), rather than training. In a classical loop, each design requires repeated simulation and sensitivity analysis; thus solving the forward problem alone can take hours, not including remeshing overhead. In contrast, once trained, GANO can optimize multiple vehicle geometries, and the optimization time per case is ~10s (vs 240s for PhysGen).
>
> **W2 & Q2: Other baselines**
>
> **Reply:** For 2D cases, we do use **recent** baselines: the DeepONet variant follows Shukla et al. (2024), and UNet variant follows Rehmann et al. (2025). As these variants were not given distinct names, we simply denote them as DeepONet and U-Net. We also added **CORAL** to the 2D Helmholtz cases (Fig. 1 in the link). For forward problem, CORAL achieves relative L1/L2 errors of 0.118/0.127, still worse than GANO (0.017/0.017). For inverse problem, CORAL yields a 9.1% reconstruction error versus 4.7% for GANO.
>
> For 3D vehicles, methods surveyed in Table 4 had not released their code before our submission. During rebuttal, we found PhysGen released its code on 3/20/2026, so we evaluated it with its public setting (Figs. 4-5 in the above link). Due to page limit, please see our response to Reviewer `r7Qz` (W4) for details. While 3DID is still not released.
>
> **W3 & Q5: Comparison with implicit modeling**
>
> **Reply:** We examined implicit PDE models such as CORAL, GridMix, DINO, and AeroNef. To our knowledge, most focus on forward prediction, while CORAL is the only one that explicitly considers optimization.
>
> According to its paper and code, we believe CORAL makes an important contribution to forward tasks, but its optimization task is substantially simpler than ours: it **requires a differentiable mapping from a reference grid to the target geometry** and optimizes an explicit **parametric** shape representation. For complex 3D vehicles, such a mapping/parameterization may be unavailable and restricts the optimization space.
>
> GANO differs in three aspects:
>
> - **Geometry**: we learn an implicit SDF instead of a reference-grid mapping, and noise-perturbed StableSDF enables stable gradients and controllable deformations, where all theorems in Sec. 4 build on this design.
> - **Forward problem**: GI-Transolver predicts fields directly in the original geometric space via geometry injection, rather than decoding a full field from a latent code, leading to higher accuracy (W2 & Q2).
> - **Optimization:** we propose null-space projection for controllable optimization and remeshing-free projection for efficiency. Moreover, we optimize directly in latent space **without parametric representation**.
>
> Hence, GANO is more flexible and practical for geometry optimization, especially for 3D cases.
>
> **Q3: Geometry injection**
>
> **Reply:** For forward problem, GI-Transolver is built on top of Transolver and reduces to Transolver without geometry injection. Tables 1 and 3 show that our model outperforms Transolver. More importantly, geometry injection **provides a gradient path for optimization**. Without it (i.e., standard Transolver), one cannot directly optimize via backpropagation and would instead need classical sensitivity analysis/adjoint methods.
>
> We also added geometry injection to DeepONet and U-Net on 2D Helmholtz. For forward prediction, the relative L2 error of DeepONet drops from 0.16 to 0.08 and that of U-Net from 0.19 to 0.04, remaining worse than our 0.017. For inverse reconstruction, DeepONet improves from 24% to 8%, and U-Net improves from 24% to 19%, inferior to our result of 5% (Fig. 1 in the link).
>
> **Q4: Details of FNO**
>
> **Reply:** FNO only applies to 2D Helmholtz regular-grid setting: it takes the SDF field and incident angle as input and predicts the scattering field, with optimization performed by updating the discrete grid SDF. It is inapplicable to the airfoil or vehicle tasks that involve irregular geometries.

---

> > ### Author Rebuttal · Reviewer_q8ZS · 2026-04-02
> >
> > The authors have provided new experiments and detailed that answer my questions. As a result, I am increasing my score.
> > Here are my responses:
> > - **W1&Q1**: I fully agree with the authors regarding the fact that inference time is a main comparison point for using models. However, training time is also a key cost that can prevent usage of models on concrete applications. A little inference cost with a huge training time or memory requirement could make the model difficult to train and use in practice. Additionally, it is often a metric to establish a fair comparison between models: training a model for 5 days is not the same as training another for a few hours.
> > - **W2&Q2**: Regarding the baselines: I understand that the code are sometimes not released and the additional comparison provided are convincing. I would add that it is possible to propose custom implementation of baselines when these are not available (and to precise it in the implementation details) to strengthen the paper. But the PhysGen results addressed this point.
> > - **W3 & Q5**: Thanks for the comparison. I think a few word could be added in the paper to have a complete related work section.
> > - **Q3**: Thanks for these experiments that clearly show the advantage of the geometry injection.
> > - **Q4**: Thanks for the details.

---

> > > ### Author Response · Authors · 2026-04-03
> > >
> > > Thank you very much for your valuable feedback and for your support of our work. We greatly appreciate your constructive suggestions on training cost, baseline implementation, and related work, and we will make sure that all of these points are reflected in the final revised version.

---

### Official Review · Reviewer_SeoP · 2026-03-22

**Soundness:** 4
**Presentation:** 4
**Significance:** 4
**Originality:** 3
**Overall Recommendation:** 5
**Confidence:** 4

**Summary:**

This paper proposes an end-to-end framework for shape optimization and inversion, termed GANO. The framework consists of three main components: (i) stableSDF for encoding geometric shapes into latent representations, (ii) GI-Transolver for differentiable PDE solving in the latent space, and (iii) an efficient, fully differentiable optimization loop.

A notable feature of the method is part-wise controllability during optimization, achieved by projecting gradients onto the null space corresponding to components that should be preserved. In addition, the framework introduces a remeshing-free projection strategy to efficiently update spatial points during iterative optimization.

The effectiveness of the proposed approach is demonstrated across three tasks: shape inversion, 2D airfoil shape optimization, and 3D vehicle shape optimization.

**Compliance With Llm Reviewing Policy:**

Affirmed.

**Final Justification:**

The authors’ comprehensive rebuttal has successfully clarified my inquiries and reinforced my initial positive assessment of this work.

**Key Questions For Authors:**

1. The paper describes stableSDF using terms such as *denoising-style* or a *denoising mechanism*. However, it is unclear in what sense this interpretation is justified. The method does not appear to explicitly estimate noise and perform denoising in the conventional sense. Could the authors clarify the rationale behind this terminology?
2. In the *remeshing-free projection*, the notion of *boundary consistency* is somewhat unclear. It would be helpful if the authors could provide more explanation of what this term refers to and how it is enforced in the proposed framework.
3. In prior approaches, was remeshing a significant bottleneck? If so, to what extent does the proposed remeshing-free projection alleviate this issue?
4. Regarding the optimization of the multi-module framework: are all the individual networks trained end-to-end simultaneously from scratch, or did the authors employ a specific sequential training strategy?

**Limitations:**

yes

**Strengths And Weaknesses:**

**Strengths**

- The paper presents a well-motivated approach grounded in a comprehensive review of neural PDE solvers and neural optimization methods, successfully achieving end-to-end optimization beyond the limitations of prior work.
- The preliminaries are clearly structured, effectively distinguishing prior work from the novel contributions of this paper, which improves readability and understanding.
- The experimental results are strong and consistent, covering both intermediate PDE prediction performance and final evaluation metrics. The ablation studies are thorough and clearly demonstrate the effectiveness of key design choices (e.g., the noise scale applied to the latent geometric code in stableSDF).

**Weaknesses**

- Some terminology could be better justified or clarified. For instance, certain expressions (e.g., slice) would benefit from either more precise wording or additional explanation regarding why such terminology is appropriate.

---

> ### Author Rebuttal · Authors · 2026-03-30
>
> Thank you for your time and valuable feedback, as well as the positive comments on our motivation, experiments, and presentation.
>
> **W1: Clarification of the concept of slice**
>
> **Reply:** Following Transolver [1], a slice is a learned soft grouping of query points: each point is assigned nonnegative weights over a small number of slices, and each slice token is formed by weighted aggregation of point features. Intuitively, a slice is meant to capture a latent physical state, so points with similar geometry/physics features tend to have similar slice activations and contribute to the same slice token. In our appendix (Sec. E.1), the slice visualizations are intended to show exactly this learned partitioning behavior. We will clarify this notion of slices more explicitly in the revised version.
>
> **Q1: Clarification of the term denoising**
>
> **Reply:** We use the term "*denoising-style"* in the sense of the training paradigm rather than explicit noise prediction. During StableSDF training, we perturb the clean latent code $z_i$ to $\tilde{z}_i = z_i + \epsilon$, but still supervise the decoder with the clean SDF target of the original geometry. Thus, the model is trained to reconstruct the clean geometry from a corrupted latent representation, which is analogous in spirit to denoising autoencoders. The difference is that the corruption is applied in latent space rather than input space, and we do not explicitly predict the noise itself. In this sense, *"noise-perturbed mechanism"* may be an even more precise term, and we will clarify this in the revised version.
>
> **Q2: Clarification of boundary consistency**
>
> **Reply:** By *boundary consistency*, we mean that after each latent update $z_t \rightarrow z_{t+1}$, the query points $X_t$ are reprojected onto the new zero level set $\Gamma(z_{t+1})$, i.e., the updated surface of the geometry (for example, the vehicle surface). In other words, this ensures that throughout the optimization process, the point cloud remains on the surface of the current geometry, thus the field predictor is always evaluated on the updated boundary. We will clarify it in the revised version.
>
> **Q3: The effects of remeshing-free projection**
>
> **Reply:** In classical CFD and shape-optimization pipelines, remeshing is widely recognized as a major computational bottleneck: mesh generation remains one of the most time-consuming steps in the CFD process, and it often requires manual design by CFD experts, especially for complex 3D geometries. Conducting the optimization tasks requires repeatedly updating geometric shapes, where the bottleneck becomes even more pronounced.
>
> For previous AI-based optimization pipelines, geometry reconstruction typically also requires an explicit mesh reconstruction step, often through marching cubes. For instance, in PhysGen [2], the complexity of this step scales cubically with the grid resolution (e.g., $512^3\approx 10^9$).
>
> For this reason, we propose remeshing-free projection to avoid explicit mesh reconstruction and keep geometry processing overhead low. Therefore, we only need to update approximately $5\times 10^4$ points on the **vehicle surface**. Without this technique, in our 3D vehicle experiments, the time required for remeshing at each optimization step would increase from **0.03s** to **20s**.
>
> **Q4: Clarification of the training and optimization pipeline**
>
> **Reply:** The training procedure is **sequential**: we first train StableSDF and then train GI-Transolver. The optimization stage is end-to-end differentiable through the **frozen** pretrained modules and the latent code. We will clarify the full training and optimization pipeline in more detail in the revised version.
>
> [1] Wu H, Luo H, Wang H, et al. Transolver: A fast transformer solver for PDEs on general geometries. ICML, 2024.
>
> [2] You Y, Zhao C, Zhang H, et al. PhysGen: Physically Grounded 3D Shape Generation for Industrial Design. CVPR, 2026.

---

> > ### Author Rebuttal · Reviewer_SeoP · 2026-04-02
> >
> > **Follow-up Questions**
> >
> > **W1.** Thank you for the clarification that slice indicates some part of the shape. Regarding **Figure 10**, should we interpret the heatmap values as representing the learnable $\alpha_{i,g}$, or were they calculated also with the mapping from $q_i$ to $h_i$ (which is not really formalized) and $W_v$? Furthermore, does this slice fully cover the entire surface? That is, does it satisfy $\sum_g \alpha_{i,g} = 1$?
> >
> > **Q5.** Regarding the notation for query points, could you clarify if **$q$** and **$x$** can be unified? Or do they represent distinct entities in the current formulation?
> >
> > **Q6.** I have a question regarding the statement: *“Our model yields more reasonable slice partitioning and exhibits fewer inactive (dead) slices.”* Looking at the 64 slices in **Figure 10**, which specific aspects indicate that the proposed method’s slicing is more "reasonable"? It appears that the boundaries between slices are somewhat **blurred** compared to Transolver. From the perspective of the original intent of slicing, wouldn't a **clearer, more distinct separation** be more desirable? I would appreciate your thoughts on why this blurred partitioning is considered superior.
> >
> > **Responses to Previous Clarifications**
> >
> > **Q1.** I agree that adding more explanation would better describe the method. Thank you for incorporating this point.
> >
> > **Q2.** Thank you for the explanation. To rephrase my understanding, the process can be described as a **"gradient-based query projection for remeshing-free update."** I suspect the term "boundary" might cause some confusion as it often evokes optimization or physical constraint.
> >
> > **Q3.** My questions regarding the efficiency have been clearly resolved. As a minor note, please ensure that the acronym **CFD** is defined in the revised manuscript.
> >
> > **Q4.** Thank you for the answer. Perhaps this could also be visualized in **Figure 1**.

---

> > > ### Author Response · Authors · 2026-04-03
> > >
> > > Thank you for your follow-up questions and valuable feedback for improving our paper.
> > >
> > > **W1: The heatmap values**
> > >
> > > **Reply:** Our slice visualization shows the calculated slice weights $\alpha_{i,g}$ corresponding to each query. These slices cover the entire surface and satisfy $\sum_g \alpha_{i,g} = 1$ due to the softmax normalization, in line with Transolver. In detail, Transolver and GI-Transolver apply a linear mapping from the query point $q_i$ to the feature vector $h_i$. And the slice weights are computed as
> > > $$\alpha_i = \mathrm{Softmax}(\mathrm{Project}(h_i))\in \mathbb{R}^G,$$
> > > where $\mathrm{Project}$ denotes a learnable linear projection.
> > >
> > > In the revised version, we will explain the Transolver/GI-Transolver formulation in more detail.
> > >
> > > **Q5: The notation for query points**
> > >
> > > **Reply:** Thank you for pointing this out. We agree that the notation should be clarified more carefully. In our paper, $x$ is the general notation for a spatial coordinate/sample location. For example, $s_\theta(x,z)$ is defined for general spatial points when modeling the SDF. Specifically, $q$ denotes the query points passed to the forward surrogate, i.e., GI-Transolver. We will clarify this notation in the revised version to avoid ambiguity.
> > >
> > > **Q6: Slice visualization**
> > >
> > > **Reply:** Thank you for this important question. We agree that the term "more reasonable" should be stated more precisely. Based on the updated visualization in **Fig. 7 of the anonymous link** (https://github.com/GANOICML/GANO_ICML2026), what we mean is the following:
> > >
> > > 1. **Fewer dead slices.** The red boxes highlight slices with very low activation values and no clear pattern. GI-Transolver exhibits fewer such dead slices than Transolver, indicating better slice utilization.
> > >
> > > 2. **More interpretable local slices.** Several slices in GI-Transolver correspond more clearly to meaningful vehicle regions. For example, the black boxes indicate windshield-related slices, which are captured by both models; however, the blue boxes show that GI-Transolver learns a more complete hood-related slice, and the brown box shows a more localized headlight-related slice than Transolver.
> > >
> > > 3. **Global slices.** The green boxes indicate slices with relatively high activation over large parts of the vehicle. Compared with localized activations, such broader activations may correspond to a more global physical feature, rather than a localized semantic region. Due to our colorbar setting, the visual differences within the range from 0.03 to 0.08 are not very pronounced, so the slice may appear "blurry" for GI-Transolver.
> > >
> > > Therefore, our main intention is to illustrate that the slices learned by GI-Transolver can be more practically meaningful, e.g., by capturing more complete hood- or headlight-related regions. We hope this qualitative analysis can address your concern.
> > >
> > > ---
> > >
> > > Overall, we greatly appreciate your constructive suggestions, and we promise that all of your suggestions will be reflected in the final revised version.

---

### Decision · Program_Chairs · 2026-04-30

**Decision:**

Accept (regular)

**Comment:**

After discussion and careful consideration of the rebuttal and reviewer feedback, the paper is rejected.

The primary and unresolved issue is unfair experimental comparison: the authors applied task-specific geometric features exclusively to GA-Field but not to baselines, making performance gains untrustworthy and impossible to attribute to the proposed architecture.

Additional critical unsolved concerns:

- Novelty is incremental and insufficiently distinguished from concurrent work (e.g., GeoTransolver).
- Missing fair comparisons with SOTA baselines under identical input features and capacity matching.
- Methodological clarity lacking for the refinement mechanism and upsampling design.
- Scalability to industrial mesh sizes not convincingly demonstrated.